# Impact of water uptake and mixing state on submicron particles deposition in the human respiratory tract (HRT): Based on explicit hygroscopicity measurements at HRT-like conditions

Ruiqi Man[1], Zhijun Wu[1,2], Taomou Zong[1], Aristeidis Voliotis[3,4], Yanting Qiu[1], Johannes Größ[5], Dominik van Pinxteren[5], Limin Zeng[1], Hartmut Herrmann[5], Alfred Wiedensohler[5], Min Hu[1]

[1]State Key Joint Laboratory of Environmental Simulation and Pollution Control, College of Environmental Sciences and Engineering, Peking University, 100871, Beijing, China

[2]Collaborative Innovation Center of Atmospheric Environment and Equipment Technology, Nanjing University of Information Science and Technology, 210044, Nanjing, China

[3]National Centre for Atmospheric Science, Department of Earth and Environmental Science, School of Natural Sciences, The University of Manchester, Oxford Road, M13 9PL, Manchester, UK

[4]Centre for Atmospheric Science, Department of Earth and Environmental Science, School of Natural Sciences, The University of Manchester, Oxford Road, M13 9PL, Manchester, UK

[5]Leibniz Institute for Tropospheric Research, 04318, Leipzig, Germany

*Correspondence to:* Zhijun Wu (zhijunwu@pku.edu.cn)

**Abstract.** The particle hygroscopicity plays a key role in determining the particle deposition in the human respiratory tract (HRT). In this study, the effects of hygroscopicity and mixing state on regional and total deposition doses based on the particle number concentration for the children, adults, and the elderly were quantified using the Multiple-Path Particle Dosimetry model based on the size-resolved particle hygroscopicity measurements at HRT-like conditions (relative humidity = 98%) performed in the North China Plain. The measured particle population with an external mixing state was dominated by hygroscopic particles (number fraction = (91.5 ± 5.7)%, mean ± standard deviation (SD), the same below). Particle hygroscopic growth in the HRT led to a reduction by around 24% in the total doses of submicron particles for all age groups. Such reduction was mainly caused by the growth of hygroscopic particles and was more pronounced in the pulmonary and tracheobronchial regions. Regardless of hygroscopicity, the elderly group had the highest total dose among three age groups, while the children received the maximum total deposition rate. With 270 nm in diameter as the boundary, the total deposition doses of particles smaller than this diameter were overestimated and those of larger particles were underestimated assuming no particle hygroscopic growth in the HRT. From the perspective of the daily variation, the deposition rates of hygroscopic particles with an average of $(2.88 \pm 0.81) \times 10^9$ #/h during the daytime were larger than those at night $((2.32 \pm 0.81) \times 10^9$ #/h). On the contrary, hydrophobic particles interpreted as freshly emitted soot and primary organic aerosols exhibited higher deposition rates at nighttime $((3.39 \pm 1.34) \times 10^8$ #/h) than those in the day $((2.58 \pm 0.76) \times 10^8$ #/h). The traffic emissions during the rush hours enhanced the deposition rate of hydrophobic particles. This work provides a more explicit assessment of the impact of

hygroscopicity and mixing state on the deposition pattern of submicron particles in the HRT.

**Keywords:** hygroscopicity; mixing state; HH-TDMA; lung deposition; MPPD

## 1 Introduction

Toxicological and epidemiological studies showed that ambient particles can result in the declining life expectancy and rising premature mortality (Chen et al., 2013; Correia et al., 2013; Pope and Dockery, 2013; Dockery et al., 1993; Pope et al., 2009). Compared with coarse particles, submicron particles (i.e., particles with diameter $\leq 1$ μm) have smaller sizes and larger specific surface areas, which tend to carry more toxic and harmful components and reach deeper into the human respiratory tract (HRT) (Oberdorster, 2001). Inhaled particles deposit along the HRT mainly by diffusion, sedimentation, impaction, and interception (Wang et al., 2018a). The major deposition mechanism depends on the particle size and specific deposition location (Varghese and Gangamma, 2009). Unlike ambient environments, conditions in the HRT are warm and humid, where the relative humidity (RH) can be as high as 99.5% (Hussein et al., 2013). The unique environment can alter the chemical and physical characteristics of inhaled particles, leading to variations in particle deposition distributions and doses. To accurately quantify the deposition pattern of submicron particles in the HRT, it is therefore critical to account for such potential transformations and characterize inhaled particle properties in the HRT.

Due to experimental limitations of measuring inhaled particle number size distributions (PNSDs), regional doses in the HRT are typically estimated by means of mathematical models (Hofmann, 2011). The most widely used dosimetry models are the International Commission on Radiological Protection (ICRP, 1994) model and the Multiple-Path Particle Dosimetry (MPPD) model (Asgharian et al., 2001). Estimating the particle deposition fraction (DF) in these models is based on aerosol properties as well as individual's physiological parameters. These available dosimetry models, however, fail to incorporate some critical particle characteristics, especially the hygroscopicity (Ferron, 1977), which may cause variations in the particle size and therefore affect the deposition efficiency and pattern of particles in different lung regions.

To date, many studies have assessed the effects of hygroscopicity on ambient particle deposition in the HRT based on assumed values of the hygroscopic parameter (Kappa, κ) representing non-hygroscopic, nearly hydrophobic, and hygroscopic particles (Voliotis and Samara, 2018) or estimations by parametric methods (Ching and Kajino, 2018; Hussein et al., 2013; Mitsakou et al., 2007; Haddrell et al., 2015; Vu et al., 2018). However, it is well-known that continental aerosols typically show an external mixing state and size-dependent hygroscopicity (Zong et al., 2021). Thus, in order to capture the real and high-time-resolution features of ambient particles' hygroscopic growth in the HRT, direct particle hygroscopic growth measurements are a matter of necessity.

To our best knowledge, there are only limited studies exploring the impact of the hygroscopic growth of ambient particles on the particle deposition by direct hygroscopicity measurements. Moreover, hygroscopicity measurements using the Humidity Tandem Differential Mobility Analyzer (H-TDMA) (Londahl et al., 2009; Farkas et al., 2022; Vu et al., 2015; Kristensson et al., 2013) or Differential Aerosol Sizing and Hygroscopicity Spectrometer Probe (DASH-SP) (Youn et al., 2016) in these studies were all conducted at relatively lower RH (~ 90%) compared to that in the HRT (RH = 99.5%). For example, Farkas et al. (2022) modelled DFs of aerosol particles with four different diameters and studied in their dry state

and after their hygroscopic growth at RH = 90% using a H-TDMA. Youn et al. (2016) examined size-resolved hygroscopicity data by DASH-SP for particles sampled near mining and smelting operations to study the effects of particles' hygroscopic growth on the HRT deposition of toxic contaminants. It was further assumed that $\kappa$ was independent of RH on the premise that the effective molar volume of the solute does not vary with RH. However, the presence of surface active, slightly soluble substances, and the co-condensation of semi-volatile soluble organic compounds can result in the humidity-dependent characteristic of $\kappa$ (Wu et al., 2013; Wex et al., 2009; Topping and Mcfiggans, 2012). For instance, Liu et al. (2018) showed that $\kappa$ could vary from about 0.1 at RH < 20% to less than 0.05 when RH $\approx$ 90% due to the non-ideal mixing of water with hydrophobic and hydrophilic organic components. Therefore, an explicit hygroscopicity measurements at HRT-like conditions will make the deposition estimation more accurate.

In this study, the size-resolved particle hygroscopicity derived from a high humidity tandem differential mobility analyzer (HH-TDMA) at HRT-like conditions (RH = 98%) was first used to quantify the effects of both hygroscopicity and external mixing state on the particle deposition in the HRT using the MPPD model. The deposition doses of submicron particles were calculated in the head, tracheobronchial (TB), and pulmonary (P) regions in the HRT for different age groups. Further, the diurnal variations of deposition rates of hygroscopic and hydrophobic particles were also calculated to provide an insight into the particle deposition linked to human activities.

## 2 Materials and Methods

### 2.1 The Sampling Site and Instruments

The field campaign was conducted from June 8 to July 6 in 2014 at an ecological park in the rural area of Wangdu County (38.666ºN, 115.210ºE) in the North China Plain, a polluted area with high population density and strong primary emissions. The surroundings were wheat fields without significant industry emissions. A detailed description of the sampling site can be found in our previous study (Wu et al., 2017b). In brief, a HH-TDMA and a twin differential mobility particle sizer (TDMPS) were employed to measure the hygroscopic growth factor (HGF) of specific size particles at RH = 98% and the size-resolved PNSDs of particles ranging from 3 to 800 nm respectively. Besides, NO and CO was monitored by a NOx chemiluminescence analyzer (42i-TLE, Thermo Scientific) and a Trace Level Carbon Monoxide Analyzer (48iQ, Thermo Scientific), respectively (Chen et al., 2020). The BC mass concentrations were measured by a Multi-Angle Absorption Photometer (MAAP Model 5012, Thermo, Inc.) (Wang et al., 2019). The OH radical was measured by a Laser-induced fluorescence (LIF) (Tan et al., 2020).

### 2.2 Particle Hygroscopic Growth Measurement

The HH-TDMA was designed to measure the aerosol hygroscopic growth at high RH (90% - 98%), using the technique of a temperature-controlled water bath, which is able to hold the RH of aerosols and sheath flow stable for RH > 90% (Hennig et al., 2005). The RH in the second differential mobility

analyzers (DMA) reached an absolute accuracy of ±1.2% for 98% and a long-term stability of ± 0.1-0.4% of set point values (Hennig et al., 2005). More detailed information regarding the HH-TDMA system was provided by Bian et al. (2014) and Wu et al. (2017a). The HGF was defined as the ratio of the wet particle diameter at a given RH ($D_{P, wet}$) to the dry particle diameter for RH < 10% ($D_{P, dry}$):

$$HGF = \frac{D_{P, wet}}{D_{P, dry}}, \tag{1}$$

The TDMAinv method developed by Gysel et al. (2009) was used to invert hygroscopicity data of settled diameter particles (30, 50, 100, 150, 200, and 250 nm) to HGFs of size-resolved particles and hygroscopic growth factor probability distribution function (GF-PDF) at RH = 98% (Gysel et al., 2009). The HGF was then converted into the hygroscopic parameter (κ) according to the κ-Köhler theory (Petters and Kreidenweis, 2007):

$$\kappa = (HGF^3 - 1)(\frac{\exp\left(\frac{A}{D_{P, dry} \cdot HGF}\right)}{RH} - 1), \tag{2}$$

$$A = \frac{4\sigma_{s/a}M_w}{RT\rho_w}, \tag{3}$$

where HGF and $D_{P, dry}$ are the hygroscopic growth factor measured at 98% RH by HH-TDMA and the dry particle diameter respectively. $\sigma_{s/a}$ is the droplet surface tension (assumed to be that of pure water, $\sigma_{s/a}$ = 0.0728 N m$^{-2}$). $M_w$ is the molecular weight of water. $\rho_w$ is the density of liquid water. R is the universal gas constant, and T is the absolute temperature.

The head region, or upper respiratory tract, includes the nasal cavities, the pharynx, and the larynx. In this study, the wet diameter of aerosols above the larynx was assumed to be equilibrated with ambient air conditions (Ching and Kajino, 2018), therefore set to the average value during the sampling period (T = 26 ℃, RH = 60%). The TB and P regions belong to the lower respiratory tract which were saturated with water vapor, and the temperature and RH inside were 37 ℃ and 99.5%, respectively (Vu et al., 2015). The wet particle diameter in the HRT was estimated by using the variant of Eq (2). Detailed information can be found in Farkas et al. (2022). It should be noticed that the inhaled particle was assumed to reach the equilibrium size immediately once particles enter into the HRT in this study. However, the hygroscopic growth rate of particles depends on the particle size, hygroscopicity, the ambient environment conditions (such as RH and temperature), and the residence time in the body (Ching and Kajino, 2018). A previous study pointed out that particles with $D_p$ = 100 nm can reach the equilibrium size in a few seconds, while the equilibration timescale of particles with $D_p$ > 1 μm turn to minutes (Ching and Kajino, 2018).

**2.3 Total, Hygroscopic, and Hydrophobic Particle Number Size Distributions**

The TDMPS includes two Hauke-type DMA that have different effective center rod lengths which measure aerosol particles of 20 - 800 nm and 3 - 20 nm, respectively. The two condensation particle counters (CPC) count particles downstream of DMAs. Combining the counts from the two CPCs, the TDMPS can measure the PNSD of particles from 3 to 800 nm (electrical mobility diameter). The

instrument principle and structure of TDMPS can be found in Birmili et al. (1999) and Wiedensohler et al.
(2012). To match the particle size range in the MPPD model, the electrical mobility diameter was
converted to aerodynamic diameter by Eq (4) (Khlystov et al., 2004):

$$d_a = d_m \sqrt{\mathcal{X} \times \frac{\rho \times C_{c(dm)}}{C_{c(da)}}}, \tag{4}$$

where $d_a$ and $d_m$ is the particle aerodynamic diameter (nm) and electrical mobility diameter (nm),
respectively. $\rho$ is the particle density (1.5 g cm$^{-3}$ in this study (Hu et al., 2012)). $\mathcal{X}$ is the shape factor. $C_c$ is
the Cunningham slip correction factor for a certain diameter. Similar to other studies, the shape factor $\mathcal{X}$ is
assumed as 1 and $C_c$ is neglected in the calculation (Khlystov et al., 2004; Hu et al., 2012). Therefore, the
electrical mobility diameter (in the range of 10.3 - 756.6 nm) was converted to the aerodynamic diameter
(in the range of 12.6 - 926.6 nm).
Taking particle mixing state into account, the particle population can be categorized into hygroscopic
and hydrophobic groups according to HGFs measured at RH = 98% by HH-TDMA. Particles with HGF <
1.2 were regarded as hydrophobic particles whereas those with HGF ≥ 1.2 were regarded as hygroscopic
particles (Wang et al., 2018b; Zong et al., 2021). The hydrophobic particles in urban environments have
been interpreted as originating from freshly emitted soot and vehicle exhaust, while the hygroscopic
particles have been regarded to experience long-distance transport (Swietlicki et al., 2008; Baltensperger,
2002). To obtain the PNSDs for both groups, the total PNSD measured by TDMPS were scaled by the
number fractions (NF) of hydrophobic and hygroscopic particles. With HGF = 1.2 as the cut-off point, the
GF-PDF for each selected size was divided into the hydrophobic and hygroscopic modes. The calculation
methods of the HGF and NF of each mode were detailed in Zong et al. (2021). The NFs of size-resolved
particles within the measuring range of the TDMPS were calculated by linear interpolation methods, while
those of particles out of the range were equal to the closest known NF.
**2.4 Particle Dose Estimation**
The MPPD model (version 3.04) was used to estimate the deposition of particles in the HRT, since it
fits with the measured data better than the ICRP model (Londahl et al., 2007). This model calculates
deposition and clearance of monodisperse and polydisperse aerosols in the size range of 1 nm - 100 µm in
the respiratory tracts of laboratory animals, human adults, and children. Within each airway, the deposition
is calculated using theoretically derived efficiencies for deposition by diffusion, sedimentation, impaction,
and interception within the airway or airway bifurcation (MPPD: Multiple-Path Particle Dosimetry Model,
2022). The model requires the following parameters as input: (1) airway morphometry parameters (airway
morphometry model, functional residual capacity (FRC), and the upper respiratory tract (URT) volume); (2)
particle properties (density, diameter); (3) exposure scenario (breathing frequency (BF), tidal volume (TV));
and (4) deposition/clearance.
In this study, the stochastic model (60$^{th}$ percentile) was chosen, which is closer to the realistic structure
of human lungs (Voliotis and Samara, 2018; Li et al., 2016; Asgharian et al., 2001; Wang et al., 2021; Lyu
et al., 2018; Avino et al., 2018; Manigrasso et al., 2015). The particle diameter range was set as 0.01 - 10.0
μm. Particle density was taken as 1.5 g/cm$^3$ according to a previous study in Beijing (Hu et al., 2012). In
order to obtain the deposition pattern of different age groups, the population was divided into three groups
on the basis of their age: children (7 - 12 years old), adults (18 - 26 years old), and the elderly (> 59 years
old). Then, the particle deposition was estimated based on Chinese localized physiological parameters
(Table 1). These values were considered for an exposure scenario for resting (e.g., sitting) and nasal
breathing. All the model simulations were conducted based on this exposure scenario using the male
physiological parameters, as corresponding data for females were not available. Elsewhere it was shown
that males received higher doses compared to females in all age classes due to the different physiological
parameters (e.g., higher TV and FRC) (Voliotis and Samara, 2018), hence similar behavior would have
been expected here. It should be noticed that the exposure time data came from the statistical results of the
questionnaire survey of the outdoor activity time for the rural population in Hebei, China. While, people
may rest, take light exercise, or take heavy exercise during the exposure time. Different exercise levels (e.g.
sitting, walking, exercising, etc) can result in different dose estimations and are not discussed here. For
instance, previous studies found that the exercise level had great impact on the minute ventilation and led to
the increasing deposition dose (Londahl et al., 2007). All the other model input parameters were set as
default. It should be noted that any clearance mechanisms were not considered in this study, hence our
results show the upper limit of exposure.
**Table 1 Physiological and breathing parameters for three age groups**

| Age Groups[a] | FRC/mL | Height[e]/cm | URT Volume[f]/mL | TV[e]/ mL | BF[e]/ min$^{-1}$ | Exposure Time/ min day$^{-1}$ |
|---|---|---|---|---|---|---|
| Children | 1330[b,c] | 139.3 | 21.91 | 630 | 22 | 96[g] |
| Adults | 3338[d] | 158.5 | 36.31 | 730 | 18 | 253[h] |
| Elderly | 3259[d] | 166.9 | 34.01 | 760 | 18 | 241[h] |

[a] Base on the available data, age groups here refer to males.
[b] Due to the lack of data, the FRC value of children is not a Chinese localization parameter.
[c] Stocks and Quanjer, 1995
[d] Cao, 2009
[e] Zhu, 2006
[f] Hart et al., 1963
[g] Duan, 2016
[h] Duan et al., 2014
The DF is the ratio of the mass/number/surface area of the deposited particles to that of inhaled
particles in a given region. The DFs of hygroscopic and hydrophobic particles with considering
hygroscopicity (hereinafter referred to as "wet particles") were obtained based on the DFs of particles
without considering hygroscopicity (hereinafter referred to as "dry particles"). More specifically, it is
considered that the DF of wet particles with $D_p$ after hygroscopic growth was equal to that of dry particles
with the same diameter ($D_p$). Therefore, the DF curve of wet particles corresponds to shifting the particle
size range of the DF curve of dry particles to the right. The daily particle number doses and deposition rates
for size-resolved particles in a specific region were calculated as follows (Voliotis and Samara, 2018),

$$Dose_i = DF_i \times PNC_i \times TV \times BF \times t \tag{5}$$

$$Rate_i = DF_i \times PNC_i \times TV \times BF \tag{6}$$

where $Dose_i$ is the deposition dose of $i$th size particle (#/day), $Rate_i$ is the deposition rate of $i$th size
particle (#/h), $DF_i$ is the deposition fraction of $i$th size channel in a specific region, $PNC_i$ is the particle
number concentration (#/cm$^3$) corresponding to the $i$th size channel, TV is the tidal volume (mL), BF is the
breathing frequency (min$^{-1}$), and $t$ is the exposure time in ambient air (min/day). The deposition dose of a
specific region was calculated by adding together doses of size-resolved particles, and the total dose was
the sum of three regional doses. The dose without considering hygroscopicity was calculated on the basis of
the dry PNSD. The dose considering hygroscopicity was the sum of particle doses of hygroscopic and
hydrophobic groups, which was respectively calculated by the PNSD of two groups.
**3 Results and Discussion**
**3.1 Particle Number Size Distributions in the human respiratory tract**
As described in Sect. 2.3, particles were categorized into hygroscopic and hydrophobic groups at RH
= 98% according to their hygroscopicity. The hygroscopicity of size-resolved particles measured by HH-
TDMA during the sampling period was displayed in Table S1. Figure 1 showed the average ambient PNSD
under dry conditions (RH < 30%) over the entire field campaign and those of hygroscopic and hydrophobic
particles in the HRT. The wet diameters of hydrophobic and hygroscopic particles in the HRT were shown
in Table S2. The average particle number concentrations (PNCs) of hygroscopic and hydrophobic particles
were $(1.76 \pm 1.64) \times 10^4$ (mean $\pm$ standard deviation (SD), the same below) and $(1.70 \pm 3.14) \times 10^3$ #/cm$^3$,
respectively. The hygroscopic particles accounted for an average of $(91.5 \pm 5.7)$ % of the total PNC and
dominated the measured aerosol population. The PNSDs of ambient aerosols during the daytime and
nighttime were shown in Figure S1.

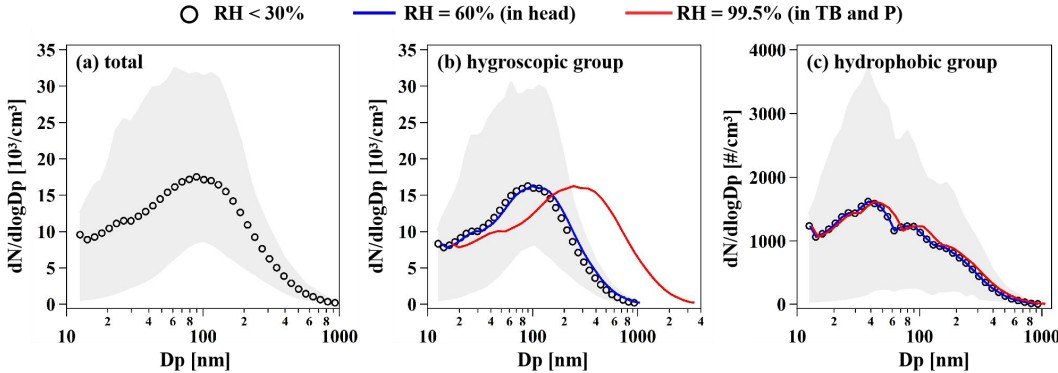


**Figure 1. (a) The average particle number size distribution (PNSD) measured by TDMPS during the sampling**
**period. The average PNSDs of the (b) hygroscopic and (c) hydrophobic groups in the human respiratory tract at**
**different relative humidities (RHs). The black markers, blue lines, and red lines represent PNSDs under dry**
As shown in Figure 1(b), the hygroscopic particles grew slightly in the head (the blue line), while they
had a remarkable growth in the TB and P regions (the red line) attributed to high humidity conditions and
water uptake. Particularly, the diameter of hygroscopic particles corresponding to the maximum PNC
shifted from about 90 nm to 250 nm. As expected, no obvious size growth of hydrophobic particles took
place in the three regions in the HRT, and the peak appeared at $D_p \approx 40$ nm (Figure 1(c)).

**3.2 Regional and Total Deposition Fractions**

Taking the adults group as an example, size-resolved regional and total DFs of particles under dry
conditions (black dots), and hydrophobic (blue dots) and hygroscopic (red dots) particles in humid
environments were shown in Figure 2. The size-resolved DFs for the children and the elderly groups were
shown in Figure S2-S3. As shown in the three figures, the regional and total DFs of all age groups
respectively followed the same trends regardless the particle hygroscopicity. Due to the similar
physiological parameters (such as the FRC, URT volume, TV, and BF, as shown in Table 1) of the adults
(Figure 2) and the elderly (Figure S3), their regional and total DF functions were similar to each other.
While, for the reason that the FRC is positively correlated with the body weight, the FRC of the children
was nearly one third of those of the adults and the elderly, which may lead to the different DF curves of the
children (Figure S2). Compared with the adults and the elderly, the children had lower DFs of ultrafine
particles in the head and higher DFs of submicron particles in the P region, resulting in higher total DFs of
submicron particles.

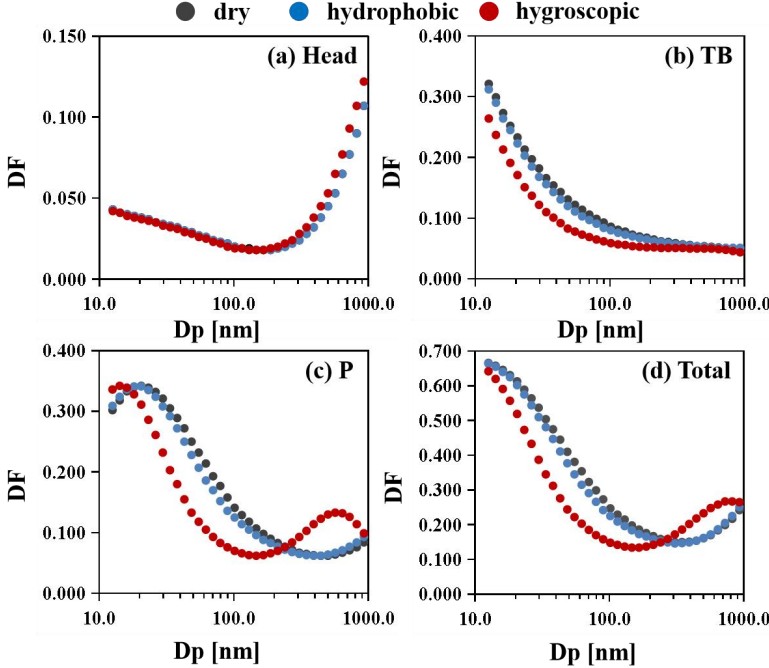

**Figure 2. Size-resolved (a) head, (b) TB, (c) P, and (d) total deposition fractions (DFs) of particles under dry**
**conditions (i.e., without considering hygroscopicity), and hydrophobic and hygroscopic particles in humid**

**256** **environments (i.e., considering hygroscopicity) for the adults group. For clarity, the abscissa of Figure 2 refers to**
**257** **the dry diameter of particles. The black, blue, and red dots represent dry, hydrophobic, and hygroscopic**
**258** **particles, respectively. In Figure 2(a), the black dots representing DFs under dry conditions is hidden behind the**
**259** **blue dots representing DFs of hydrophobic particles, because these two sets of DFs are close to each other.**

**260** As shown in Figure 2(a), there was no significant difference between head DFs whether
**261** hygroscopicity was considered or not, because the conditions within the upper respiratory tract were in
**262** balance with the surrounding environment (Ching and Kajino, 2018). With the increasing $D_p$, the head DFs
**263** decreased at first and reached the minimum at around 150 nm, then increased sharply, which is similar to
**264** the trend of the head DF curve using MPPD by Youn et al. (2016).

**265** In the TB, the DF declined monotonically associated with the increasing particle diameter, and tended
**266** to plateau for particles with $D_p > \sim 100$ nm (Figure 2(b)), which is consistent with the results of previous
**267** studies (Youn et al., 2016; Hussein et al., 2019; Varghese and Gangamma, 2009). If hygroscopicity was not
**268** taken into consideration, DFs of submicron particles were overestimated for both groups (i.e., the blue dots
**269** for hydrophobic particles and the red dots for hygroscopic particles). Using the percentage of the difference
**270** between DFs in two cases (i.e., considering hygroscopicity or not) and the DF without considering
**271** hygroscopicity as a weight, tracheobronchial DFs of hydrophobic particles were overestimated by less than
**272** 9.7%, while, 23.0% on average for hygroscopic particles. For example, the DF of particles with dry
**273** diameters at around 50 nm was 0.131 without considering hygroscopicity, and it shifted to 0.120 for
**274** hydrophobic and 0.083 for hygroscopic particles considering hygroscopicity, with the differences of 8.4%
**275** and 36.6%.

**276** Compared with size-resolved DFs of particles under dry conditions (the black dots in Figure 2(c)), the
**277** DFs of hydrophobic (the blue dots) and hygroscopic (the red dots) particles in the P region were
**278** overestimated in the range of 20 - 500 nm and 20 - 250 nm respectively, and those of particles outside the
**279** above diameter ranges were underestimated. The DFs of hydrophobic and hygroscopic particles were
**280** respectively overestimated (underestimated) by up to 14.1% (10.7%) and 53.1% (109.7%). Similarly,
**281** taking particles with a dry particle diameter of 50 nm as an example, the DF in the P was 0.250 without
**282** considering hygroscopicity, while it reduced to 0.133 (0.228) for hygroscopic (hydrophobic) particles
**283** considering hygroscopicity. Besides, resembled with the results of Youn et al. (2016) and Varghese et al.
**284** (2009), there was only one peak at the DF curve of the P region in the submicron range without considering
**285** hygroscopicity (black dots). Considering hygroscopicity, another peak of larger hygroscopic particles ($\sim$
**286** 800 nm) appeared. It indicates that particle hygroscopicity enables more submicron particles with relatively
**287** large diameters to deposit in the deepest parts of the lung. It should be noted that the particle density would
**288** change during hygroscopic growth, which was not considered in the calculation due to the lack of the
**289** measurement of the particle density. The sensitivity analysis of the particle density on the regional DFs was
**290** shown in Figure S4.

**291** In Figure 2(d), the total DFs of hydrophobic particles (the blue dots) had a similar trend as those of dry
**292** particles due to regional DFs mentioned above. For the hygroscopic group (the red dots), with $D_p = 270$ nm
**293** as the boundary, DFs of smaller particles were overestimated by 27.6% on average, while those of larger

particles were underestimated by 28.6% on average. When considering hygroscopicity, submicron particles undergo hygroscopic growth by water uptake, and particle sizes increase as a whole (Figure 1(b) and (c)). For small particles dominated by diffusion, as $D_p$ increasing, Brownian motion intensity decreased, and the diffusion deposition decreased accordingly. For large particles dominated by interception and inertial impaction, these two efficiencies increased with the particle size. Therefore, the corresponding particle deposition increased. This result is consistent with the previous study using the ICRP model which concluded that the deposition of particles with $D_p$ < 200 nm was overestimated without considering water uptake (Vu et al., 2015). Additionally, the trend of the total DFs in both cases in this study is similar to the experimental data of breathing NaCl with/without hygroscopicity through noses (Chalvatzaki and Lazaridis, 2018). In general, hygroscopicity has a significant effect on regional DFs of hygroscopic particles in the TB and P regions. While, no obvious variation occurred in the DFs of two groups in the head or those of hydrophodic particles in the TB and P.

**3.3 Regional and Total Deposition Doses and Rates for Different Age Groups**

Regional (head, TB, and P) and total deposition number doses with/without considering hygroscopicity were calculated for the children (a), adults (b), and the elderly (c) in Figure 3. Specific values of deposition doses of hydrophobic and hygroscopic particles in two cases can be found in Table S3-S5. Hussein et al. (2013) found that the deposited dose calculations in the other age groups (the elderly and teens) were in the same order of magnitudes as that of the adults. This is also true in our results. In both cases, the elderly group had the highest total deposition dose among the three groups, followed by the adults and children. While, Voliotis et al. (2018) concluded that the adults received the highest doses among all age groups, which may be caused by different physiological parameter values, such as the TV. In each group, the contribution of the P region to the total dose was the greatest ($> \sim 55\%$), which was similar to the published conclusions (Voliotis and Samara, 2018; Hussein et al., 2013; Manigrasso et al., 2017; Li et al., 2016). Moreover, the proportion of pulmonary deposition in total doses for the children was up to 62.9% considering hygroscopicity. By comparison, it accounted for 54.6% and 55.7% for the adults and the elderly groups, respectively. This indicates that particles inhaled by children are more likely to deposit in their pulmonary regions, which is in accordance with the results of previous studies (Voliotis and Samara, 2018; Voliotis et al., 2021). Taking hygroscopicity into consideration, total deposition doses significantly reduced by about a quarter (24.0% - 24.1%) for all age groups. The greatest reduction took place in doses in the P region (25.9% - 26.3%), followed by doses in the TB (24.2% - 26.1%). Head deposition had only minor variations (-0.9% - +0.5%) in both cases.

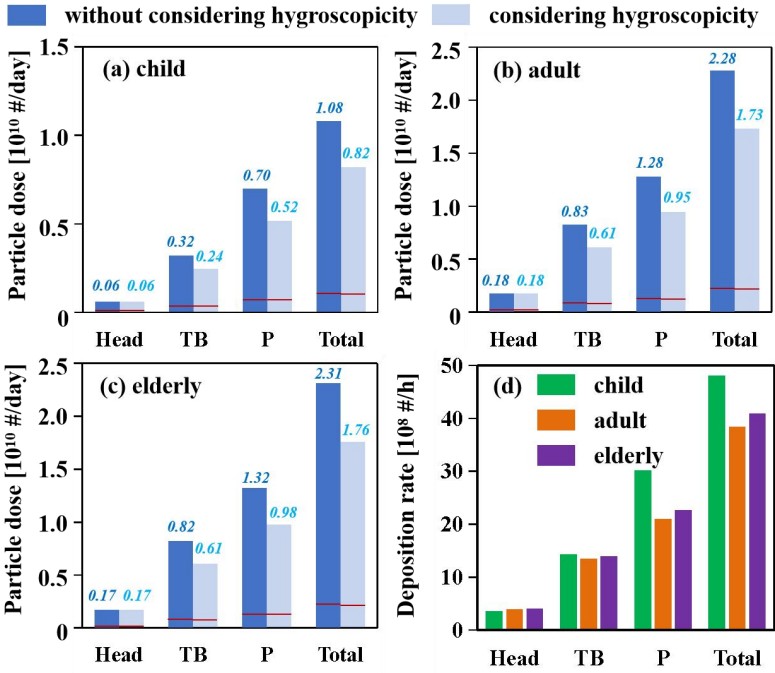

**Figure 3. Regional and total deposition doses for the (a) children, (b) adults, and (c) the elderly with/without considering particle hygroscopicity. The dark blue columns represent doses without considering hygroscopicity. The light blue columns represent doses considering hygroscopicity. The red lines on the column represent the division of doses of hygroscopic (above the red line) and hydrophobic particles (below the red line). Numbers above each column mean the corresponding particle doses with a unit of $10^{10}$ #/day. (d) The average regional and total deposition rates considering hygroscopicity for three age groups. The green, orange, and purple column represent the children, adults, and elderly, respectively.**

In both cases, the adults (Figure 3(b)) and the elderly (Figure 3(c)) groups received similar regional and total doses. In contrast, the children had the minimum total dose (Figure 3(a)), which was around half (47.4% on average) to that for the adults. As shown in Eq (4), the exposure time is an important parameter for deposition dose calculations. The exposure time of the adults and the elderly groups was more than twice than that of the children (Table 1), which resulted in the greater deposition dose for the former two groups. Therefore, to remove the impact of the exposure time, the regional and total deposition rates for three age groups were also calculated and shown in Figure 3(d). The children received the maximum total deposition rate (($4.81 \pm 4.55$) × $10^9$ #/h), followed by the elderly group (($4.09 \pm 3.92$) × $10^9$ #/h), and the adults received the minimum (($3.84 \pm 3.69$) × $10^9$ #/h). The regional deposition rate in the TB and P regions for three age groups showed a same order as the total deposition rate, while the order in the head was quite different. Specifically, three age groups had the similar deposition rate in the head.

As shown by the red lines in Figure 3, the contribution of hydrophobic particles to total deposition doses was about 10.0% for all age groups, while it increased to 12.5% after considering hygroscopicity. Hydrophobic particles were assumed to originate from freshly emitted soot and exhaust particles (Swietlicki et al., 2008; Baltensperger, 2002), which are composed of species that do great harm to human health, such as black carbon (BC) (Highwood and Kinnersley, 2006), primary organic aerosols (Mauderly and Chow, 2008), and polycyclic aromatic hydrocarbons (Kim et al., 2013; Haritash and Kaushik, 2009).

Therefore, we need to pay attention to the deposition effects of hydrophobic particles.

The adults' regional and total deposition doses of size-resolved particles with/without considering

hygroscopicity were shown in Figure 4. The effects of hygroscopicity in particle doses in the head (Figure
4(a)) are insignificant, because the RH in the upper respiratory tract was close to that under dry conditions.
Regional doses in the TB decreased due to particle hygroscopic growth (Figure 4(b)), and the greatest
reduction (~ 33%) appeared between 40 and 80 nm in diameter. Similarly, particle hygroscopicity
considerably decreased deposition doses (up to 50.3%) in the P region for particle sizes between 20 and 240
nm (Figure 4(c)). Inversely, particle doses increased (up to 102.6%) in the P region for diameters less than
20 nm and above 240 nm due to hygroscopic growth. As a result, the total deposition dose, as shown in
Figure 4(d), was overestimated for particles smaller than around 270 nm with a maximum of 40.8%
without considering hygroscopicity. The deposition doses of particles larger than this diameter were
underestimated and the maximum was 43.0%.

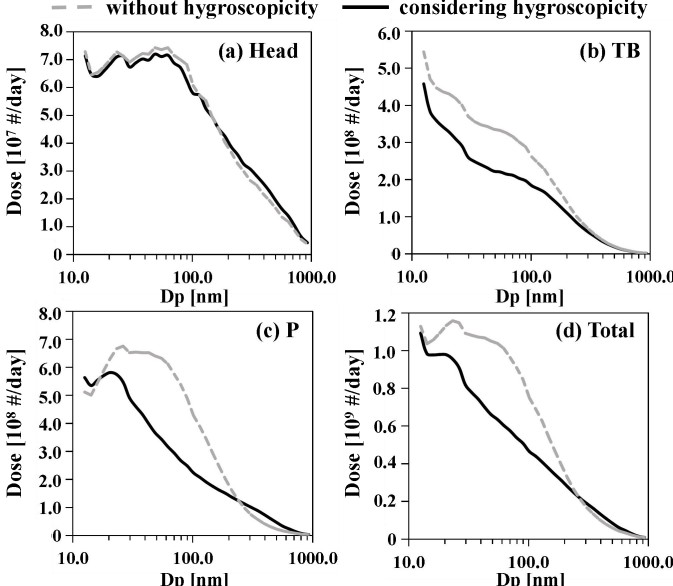


**Figure 4. (a) Head, (b) TB, (c) P, and (d) total deposition doses of size-resolved particles for the adults group**
**with/without considering hygroscopicity. For clarity, the abscissa of Figure 4 refers to the dry diameter of**
**particles. The grey dashed line represents doses without considering hygroscopicity. The black solid line**
**represents doses considering hygroscopicity.**
**3.4 Deposition Rates of Hygroscopic and Hydrophobic Particles**

In order to link human daily activities to the particulate matter deposition, the diurnal variations of

deposition rates of hygroscopic and hydrophobic particles considering hygroscopicity for the adults group
averaged over the entire field campaign were investigated and displayed in Figure 5. Additionally, the
average concentrations of NO, CO, BC, and OH radical were also given in Figure 5. By comparison, the
diurnal variations of the deposition rates of hygroscopic and hydrophobic particles without considering
hygroscopicity were displayed in Figure S5. No matter which time was considered during a day, the
deposition rate of hygroscopic particles $((3.60 \pm 6.68) \times 10^9$ #/h) in the HRT was nearly one magnitude
higher than that of hydrophobic particles $((5.15 \pm 14.4) \times 10^8$ #/h).

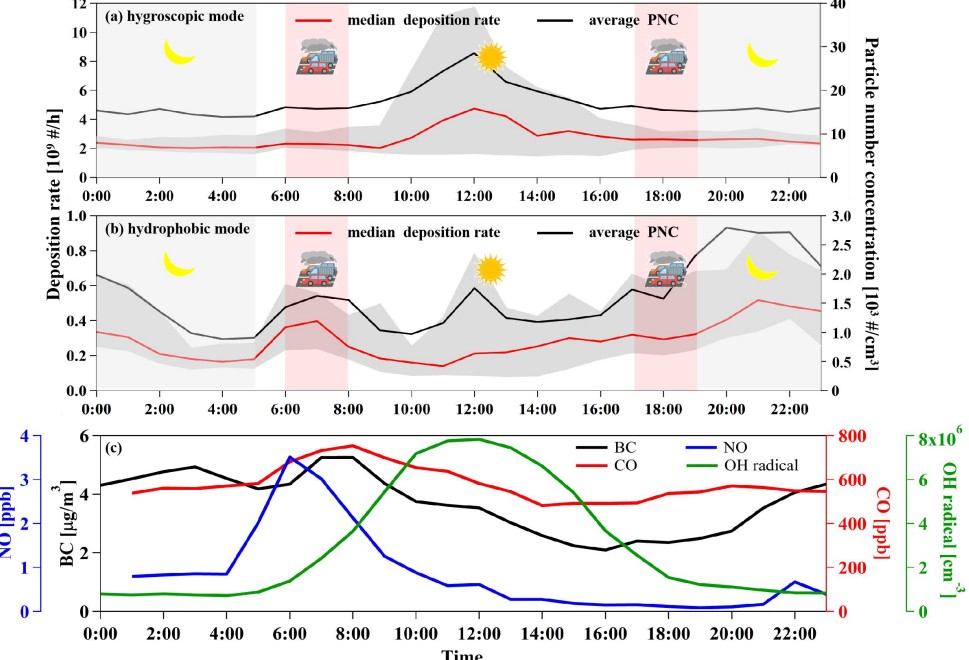


**Figure 5. The diurnal variations of deposition rates considering hygroscopicity and particle number concentrations (PNCs) of (a) hygroscopic and (b) hydrophobic particles for the adults group. The red lines represent the median deposition rate. The upper and lower edges of the grey area represent the 75th and 25th quantiles of deposition rates, respectively. The black lines represent the average PNC. (c) The diurnal variations of the average concentrations of NO, CO, BC, and OH radical during the sampling period. The blue, red, black, and green lines represent NO, CO, BC, and OH radical, respectively.**

The deposition rate of hygroscopic particles (Figure 5 (a)) was higher during the daytime (5:00 - 19:00, $(2.88 \times 10^9) \pm (8.10 \times 10^8)$ #/h) than that at night (from 20:00 to 4:00 the next day, $(2.32 \times 10^9) \pm (2.41 \times 10^8)$ #/h). The peak appeared at noon and the deposition rate reached the top at 12:00 ($4.74 \times 10^9$ #/h on average). The enhanced deposition rate can be attributed to the strong atmospheric oxidation capacity during the daytime, indicated by OH radical in Figure 5(c), which creates new particles or transfers the pre-existing aerosols to more aged particles through nucleation, semi-volatile partitioning, and multiphase chemistry (Raes et al., 2000; Donahue et al., 2014; Tan et al., 2020; Rudich et al., 2007). For instance, a previous study pointed out that the atmospheric oxidation capacity may enhance the particle formation by oxidizing the trace gases emitted from sources (such as biomass burning) and producing low-volatility condensable vapors, which nucleates in the plume (Wu et al., 2017b). Therefore, the ultrafine particles may be produced during transport (Wu et al., 2017b). During the field campaign, the new particle formation (NPF) events took place frequently (Figure S6). Our previous study showed that the NPF events and subsequent growth produced a large amount of hygroscopic and internally mixed particles (Wu et al., 2017b), thus increasing the PNC (as shown in the black line in Figure 5(a)) and leading to the enhanced deposition rate of hygroscopic particles in the day. It is indicated that the deposition rates of particles were mainly influenced by the PNC.

On the contrary, hydrophobic particles (Figure 5(b)) exhibited higher deposition rate at nighttime

$((3.39 \pm 1.34) \times 10^8$ #/h) than that in the day $((2.58 \times 10^8) \pm (7.60 \times 10^7)$ #/h). The deposition rate of

hydrophobic particles peaked at 6:00 - 8:00 during morning rush hours $(3.37 \times 10^8$ #/h on average), as

indicated by NO, CO, and BC concentrations in Figure 5(c). In the evening, the deposition rate step-wisely

increased and reached the maximum $(5.17 \times 10^8$ #/h on average) at around 21 o'clock. Correspondingly,

BC concentrations increased as well. The strong primary emissions (such as the biomass burning process

(Wu et al., 2017a)), weak chemical processes, and low boundary layer height (Zhang et al., 2014) resulted

in the increased hydrophobic particle number concentration. Thus, people who are exposed to outdoor air

during rush hours and the evening may have a higher exposure risk to hydrophobic particles.

**4 Conclusions and Implications**

To accurately quantify the effects of both hygroscopicity and external mixing state on the particle

deposition, the size-resolved particle hygroscopicity measured at HRT-like conditions (RH = 98%) was

used to estimate the deposition doses of submicron particles in the HRT for different age groups

with/without considering hygroscopicity using the MPPD model.

The total particle number concentrations were dominated by the hygroscopic particles (number

fraction = $(91.5 \pm 5.7)$%). Taking hygroscopicity into consideration, total deposition doses significantly

reduced by about a quarter (24.0% - 24.1%) for all age groups. The greatest reduction took place in doses in

the P (25.9% - 26.3%) and TB (24.2% - 26.1%) regions. Head deposition had only minor variations (-0.9%

- +0.5%) in both cases. With 270-nm as the boundary, the total doses of smaller particles were

overestimated and those of larger particles were underestimated. Regardless of hygroscopicity, the elderly

groups received the highest total doses, and the children had the lowest doses, which was around half to

that for the elderly. Pulmonary doses dominated the deposition pattern. When it comes to the deposition

rate, the children received the maximum total deposition rate, followed by the elderly group, and the adults

received the minimum. The diurnal variations of deposition rates of hygroscopic and hydrophobic particles

were also calculated. The deposition rate of hygroscopic particles was higher during the daytime $(2.88 \times$

$10^9) \pm (8.10 \times 10^8$ #/h vs. $(2.32 \times 10^9) \pm (2.41 \times 10^8)$ #/h at night) attributed to the strong atmospheric

oxidation capacity. Hydrophobic particles exhibited higher deposition rate at nighttime $((3.39 \pm 1.34) \times 10^8$

#/h) than those in the day $((2.58 \times 10^8) \pm (7.60 \times 10^7)$ #/h), which was associated with strong primary

emissions, weak chemical processes, and low boundary layer height. The traffic emissions during the rush

428       hours also enhanced the deposition rate of hydrophobic particles. Based on a more explicit hygroscopicity

measurement at RH = 98%, this work provides an insight into the impact of hygroscopicity on the

deposition pattern of submicron particles in the HRT.

Although the particle hygroscopicity was measured at a quite high RH (98%), there is still a gap

between the measurement conditions and the real HRT-conditions (RH = 99.5%). The higher RH in the

lower respiratory tract will lead to greater hygroscopic growth of particles. Since the particle

hygroscopicity may reduce the regional and total deposition doses, it is inferred that the higher RH in the

lower HRT will further decrease the deposition dose. Therefore, developing the advanced technology for
measuring the hygroscopicity in the real HRT-conditions will help to explore the influence of
hygroscopicity on the particle deposition in the HRT more accurately. For instance, the HH-TDMA (Suda
and Petters, 2013), the Leipzig Aerosol Cloud Interaction Simulator (Stratmann et al., 2004), the inverted
streamwise-gradient cloud condensation nuclei counter (Ruehl et al., 2010), and the filter-based differential
hygroscopicity analyzer (Mikhailov et al., 2011) have been used to determine the particle hygroscopicity at
RHs up to 99%. In particular, Mikhailov and Sergey (2020) adopted a new method with in situ
restructuring to minimize the influence of particle shape, and the RH was up to 99.6% with an RH
measurement accuracy better than 0.4%.

Due to the limited measurements and physiological parameters, some vital factors which may have

effect on the particle deposition in the HRT were not considered in this study, such as gender, the exercise
level, and the particle density. Besides, the deposition pattern of particles with diameters larger than 1 μm
was not discussed here due to the lack of the measurement of the PNSD and hygroscopicity of coarse
particles. While, as an important part of the ambient particle mass, coarse particles may also make a
significant contribution to the particle deposition in the HRT (Figure S7). The related research to find out
the impact of the particle hygroscopicity on the deposition mass dose of coarse particles ought to be carried
out in the future.

*Data availability.* The data presented in this article can be accessed on the open accessible research data
repository.
*Author contribution.* ZW, JG, DP, LZ, HH, and AW carried out the field observation and obtained data.
RM,TZ, and YQ processed and analyzed data. All authors discussed the results and contributed to the
writing of this paper. RM prepared the manuscript with the contributions of all co-authors. ZW, AV, and
MH further modified and improved the manuscript.
*Competing interests.* The authors declare that they have no conflict of interest.
*Acknowledgements.* We gratefully acknowledge Applied Research Associates, Inc. (ARA), The Hamner
Institutes for Health Sciences, the National Institute of Public Health and the Environment (RIVM), and the
Ministry of Housing, Spatial Planning and the Environment for developing and freely distributing the
MPPD software.
*Financial support.* This work was supported by National Natural Science Foundation of China (NSFC,
grant no. 42011530121, 91844301) .

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
