# Peer review of "Impact of water uptake and mixing state on submicron"

_EGUsphere, 2022_

## Author Comment (AC1)

**Response to reviewers**

Thanks for reviewers' careful reading and constructive comments and suggestions. We made every effort to respond to reviewers' questions point to point, and revised our manuscript and appendix according to their comments. For clarity, reviewers' comments are shown in *black italic font*. The response is shown in blue normal font. The modified content in the manuscript and/or the appendix is shown in **green bold font**.

**Anonymous Referee #1**

Referee comment on "Impact of water uptake and mixing state on submicron particles deposition in the human respiratory tract (HRT): Based on explicit hygroscopicity measurements at HRT-like conditions" by Ruiqi Man et al., EGUsphere, https://doi.org/10.5194/egusphere-2022-256-RC1, 2022

*Summary:*

*The study "Impact of water uptake and mixing state on submicron particles deposition in the human respiratory tract (HRT): Based on explicit hygroscopicity measurements at HRT-like conditions" by Man et al., examines the impact of considering hygroscopic properties of ambient aerosol into the deposition rates of particles in the HRT. It therefore uses measurements of particle size distribution (PSD) for hydrophobic and hygroscopic particles performed in the North China Plain at a RH of 98 %. The study shows the impact of a changed PSD due to hygroscopic growth of ambient aerosol. The results show the sensitivity on aerosol deposition in the HRT to PSD with a decrease in deposition around24 % when considering hygroscopic growth of the aerosols in the HRT (with large effect for hygroscopic particles and rather little for hydrophobic).*

*The study addresses an important topic and provides a promising analysis based on measurement data. In the current version of the manuscript there are no substantially new findings, however, the combination of measurements of size resolved hygroscopic and hydrophobic aerosol with HRT deposition modeling is a nice and interesting study. E.g. Ching and Kajino (2018) show comparable results for hygroscopic changes in the PSD.The study will profit from a more detailed comparison to previous studies, both on ambient aerosol properties and the HRT deposition. This comparison will also allow the authors to clearly define the novelty of their study. Furthermore, the study will benefit from some careful proofreading. Nevertheless, the study is in my opinion worth to be published after addressing the below listed points.*

*General comments:*

*1. A sensitivity analysis on the different deposition mechanisms would be helpful. E.g.,*

*by turning off single deposition mechanism in the model to estimate the relative importance of the shift to larger sizes in the PSD due to hygroscopic growth. Also, an extension of the discussion of the results taking the different deposition mechanisms into account will be helpful.*

[Response]: Thanks for the reviewer's suggestion.In the MPPD model, the deposition is calculated using theoretically derived efficiencies for deposition by diffusion, sedimentation, impaction, and interception within the airway or airway bifurcation (MPPD: Multiple-Path Particle Dosimetry Model, 2022). As far as I know, the MPPD do not offer the option to turn off any single deposition mechanism at a time. Therefore, unfortunately this suggestion cannot be achieved.

*2. Are there differences between the ICRP and the MPPD models? Why the authors chose the MPPD model rather than ICRP?*

[Response]: (1) Yes. A key difference between the ICRP and MPPD models is that the ICRP model takes the inhalable fraction of particles into consideration, while the MPPD model assumes that surrounding ambient aerosols are all inhaled into human bodies (Youn et al., 2016).

(2) The reason that the MPPD model was used in this study is that compared with the ICRP model, the estimated deposition by the MPPD model fits better with the measured data (Londahl et al., 2007). For example, the deposition of particles with sub-200 nm diameter resembles that of particles measured with the RESPI instrument of hydrophobic particles during spontaneous breathing (Kristensson et al., 2013).

**[Revise]: The content was modified in Line 170-171 in the manuscript:**

**"The MPPD model (version 3.04) was used to estimate the deposition of particles in the HRT, since it fits with the measured data better than the ICRP model (Londahl et al., 2007)."**

*3. How does the K parameter change between previous measurements and studies with lower RH and this study with RH = 98 %. How do the results compare to the previous studies with lower RH measurements? What is the expected impact of the difference between the measurements at 98 % RH and the actual nearly saturated conditions in the lower HRT?*

[Response]: (1) The hygroscopicity parameter κ depends on the chemical composition of particles. Therefore, only the comparison of κ measured simultaneously at the same sampling site makes sense. In this study, the particle hygroscopicity was only measured at RH = 98%. Besides, there has been no study comparing the κ measured by H-TDMA and HH-TDMA at the same site so far. Therefore, the comparison of κ at different RHs cannot be achieved.

(2) Compared to the hygroscopicity measurement at RH = 98%, the higher actual RH in the lower respiratory tract will lead to greater hygroscopic growth of particles. As shown in Figure 3, the particle hygroscopicity reduced the regional and total deposition doses. Therefore, it is inferred that the higher hygroscopicity in the lower

HRT will further decrease the deposition dose.

**[Revise]: The content was added in Line 445-451 in the manuscript:**

**"Although the particle hygroscopicity was measured at a quite high RH (98%), there is still a gap between the measurement conditions and the real HRT-conditions (RH = 99.5%). The higher RH in the lower respiratory tract will lead to greater hygroscopic growth of particles. Since the particle hygroscopicity may reduce the regional and total deposition doses, it is inferred that the higher RH in the lower HRT will further decrease the deposition dose. Therefore, developing the advanced technology for measuring the hygroscopicity in the real HRT-conditions will help to explore the influence of hygroscopicity on the particle deposition in the HRT more accurately."**

*4. The PSD measurements are performed to an upper limit of 800 nm, but Fig. 1 shows values until 1 μm, are these a result of a fit?*

[Response]: Yes, the particle size range in Figure 1 fitted with that measured by TDMPS. Based on the measuring principle of the TDMPS, the particle diameter of the PNSD refers to the electrical mobility diameter. While, the particle diameter in the MPPD model refers to the aerodynamic diameter. Therefore, the electrical mobility diameter was converted to the aerodynamic diameter. The detailed process was shown as follows.

**[Revise] The related statement was added in the manuscript in Line 148-155:**

**"To match the particle size range in the MPPD model, the electrical mobility diameter was converted to aerodynamic diameter by Eq (4) (Khlystov et al., 2004):**

$$d_a = d_m \sqrt{\mathcal{X} \times \frac{\rho \times C_{c(dm)}}{C_{c(da)}}} \tag{4}$$

**where $d_a$ and $d_m$ is the particle aerodynamic diameter (nm) and electrical mobility diameter (nm), respectively. ρ is the particle density (1.5 g cm$^{-3}$ in this study (Hu et al., 2012)). $\mathcal{X}$ is the shape factor. $C_c$ is the Cunningham slip correction factor for a certain diameter. Similar to other studies, the shape factor $\mathcal{X}$ is assumed as 1 and $C_c$ is neglected in the calculation (Khlystov et al., 2004; Hu et al., 2012). Therefore, the electrical mobility diameter (in the range of 10.3 - 756.6 nm) was converted to the aerodynamic diameter (in the range of 12.6 - 926.6 nm)."**

*5. May I overread this information, but what hygroscopicity is assumed for the calculation of the adjusted PSDs (Fig. 1 and Fig. 5)?*

[Response]: The hygroscopicity in Figure 1 and Figure 5 was obtained by HH-TDMA in the field measurement under real atmospheric conditions. The average κ of size-resolved particles during the entire sampling period was added in the appendix.

**[Revise]: The content was added in Line 228-229 in the manuscript:**

**"The hygroscopicity of size-resolved particles measured by HH-TDMA during the sampling period was displayed in Table S1."**

The kappa of size-resolved particles during the entire sampling period was shown in Table S1 in Line 19-20 in the appendix:

**Table S1 Kappa of size-resolved particles during the entire sampling period**

| Particle diameter (nm) | Kappa (mean ± SD) |
| --- | --- |
| 30 | 0.24 ± 0.09 |
| 50 | 0.24 ± 0.07 |
| 100 | 0.27 ± 0.06 |
| 150 | 0.28 ± 0.07 |
| 200 | 0.30 ± 0.08 |
| 250 | 0.32 ± 0.10 |

*6. The deposition dose which is one important quantity in this manuscript is highly sensitive to the exposure time. Since children are associated to a shorter exposure time, the lower dose seems to be explained by that simple and rather arbitrary quantity. The deposition rate however, is not influenced by this quantity and thus may serve better to compare the health impacts of sub-micrometer particulate air pollution. The exposure time seems quite high in my eyes (for adults and elderly around 4 hours of "resting" outdoors.*

[Response]: Thanks for the reviewer's comment. We think the deposition dose may be more intuitive as a measure of the particle deposition than the deposition rate, for the reason that it takes the human daily activities into account. The exposure time data came from the statistical results of questionnaire survey of outdoor activity time for rural population in Hebei, China. People may rest, take light exercise, or take heavy exercise in the exposure time. However, due to the availability of data, only localized physiological parameters for resting were obtained. Since the higher exercise level contributes to the greater particle deposition (Londahl et al., 2007), the actual deposition dose may be underestimated. Therefore, to remove the impact of the exposure time parameter, the regional and total deposition rates for three age groups were also calculated and added as follows.

**[Revise]: The statement was added in Line 333-341 in the manuscript:**

**"As shown in Eq (4), the exposure time is an important parameter for deposition dose calculations. The exposure time of the adults and the elderly groups was more than twice than that of children (Table 1), which resulted in the greater deposition dose for the former two groups. However, due to the availability of data, only localized physiological parameters for resting were obtained. People may rest, take light exercise, or take heavy exercise in the exposure time. Since the higher exercise level contributes to the greater particle deposition (Londahl et al., 2007), the actual deposition dose may be underestimated. Therefore, to remove the impact of the exposure time, the regional and total deposition rates**

for three age groups were also calculated and shown in Figure S5."

**Figure S5 and the related content were added in Line 69-76 in the appendix:**

"As shown in Figure S5, children received the maximum total deposition rate ((4.81 ± 4.55) × 10⁹ #/h), followed by the elderly group ((4.09 ± 3.92) × 10⁹ #/h), and the adults received the minimum ((3.84 ± 3.69) × 10⁹ #/h). The regional deposition rate in the TB and P regions for three age groups showed a same order as the total deposition rate, while the order in the head was quite different. More specifically, the three age groups had the similar deposition rate in the head."

[Figure]

**Figure S 5. The average regional and total deposition rates with considering hygroscopicity for three age groups. The green, orange, and purple column represent children, adults, and the elderly, respectively.**

*7. Is the time of hygroscopic growth considered in the MPPD modeling? Ching and Kajino (2018) pointed the importance of time in the hydration process of particles in the HRT out. The timescale until the particles reach their equilibrium size depending on RH can range in the scale of few seconds (Chuang 2003) and thus may play an important role in the upper respiratory tract.*

[Response]: Thanks for the reviewer's comment. The time of hygroscopic growth was not included in the MPPD model, and we didn't take it into account either. Therefore, we added some statement to point out this factor which may influence the deposition result.

**[Revise]: The content was added in Line 135-139 in the manuscript:**

"It should be noticed that the inhaled particle was assumed to reach the equilibrium size immediately once particles enter into the HRT in this study. However, the hygroscopic growth rate of particles depends on the particle size, hygroscopicity, the ambient environment conditions (such as RH and temperature), and the residence time in the body (Ching and Kajino, 2018). A

**previous study pointed out that particles with $D_p$ = 100 nm can reach the equilibrium size in a few seconds, while the equilibration timescale of particles with $D_p$ > 1 μm turn to minutes (Ching and Kajino, 2018)."**

*8. Are in Fig. 5 shown deposition rates for "resting" (table 1) only? It would be interesting to combine different human behaviors with the ambient pollution (e.g. physical exercise in the evening hours when pollution is high).*

[Response]: (1) Yes. The deposition calculations in this study were all based on the exposure scenario for resting.

(2) Thanks the reviewer's suggestion. There are some studies that takes human activities into consideration in the process of calculating the particle deposition (Londahl et al., 2007; Salma et al., 2015; Bennett et al., 2008; Daigle et al., 2003). Please refer to Line 192-193 in the manuscript: "Similarly, different exposure scenarios (e.g. sleeping, exercising, walking, etc) can result in different dose estimations and are not discussed here". For instance, a study found that the exercise level had little influence on the DF, but it had great impact on the minute ventilation (up to fourfold), leading to the increasing deposition dose (Londahl et al., 2007). However, our study force on the effect of the hygroscopicity on the particle deposition in the human respiratory tract, so we took a simple exposure scenario (resting) as an example. Therefore, although the exercise level is an important impact factor in the particle deposition, it was not discussed in this study. Some statement was added as follows.

**[Revise]: (1) The content was modified in Line 187-189 in the manuscript:**

**"These values were considered for an exposure scenario for resting (e.g., sitting) and nasal breathing. All the model simulations were conducted based on this exposure scenario using the male physiological parameters, as corresponding data for females were not available."**

**(2) The statement was added in Line 193-195 and 452-454 in the manuscript:**

**"For instance, previous studies found that the exercise level had great impact on the minute ventilation and led to the increasing deposition dose (Londahl et al., 2007)."**

**"Due to the limited measurements and physiological parameters, some vital factors which may have effect on the particle deposition in the HRT were not considered in this study, such as gender, the exercise level, and the particle density."**

*Specific comments:*

*1.   Fig.1: The figure would profit from including the range of the PSD. Also in the appendix a comparable figure for day and nighttime would be interesting for better understand the ambient aerosol conditions.*

[Response]: According to the reviewer's suggestion, Figure 1 was modified and the PNSDs during the daytime and nighttime were shown in the appendix.

[Revise]: Figure 1 was shown in Line 237-243 in the manuscript:

[Figure]

Figure 1. (a) The average particle number size distribution (PNSD) measured by TDMPS during the sampling period. The average PNSDs of the (b) hygroscopic and (c) hydrophobic groups in the human respiratory tract at different relative humidities (RHs). The black markers, blue lines, and red lines represent PNSDs under dry conditions (RH < 30%), in the head (RH = 60%), and in the TB and P (RH = 99.5%), respectively. The upper and lower edges of the grey area represent the 90th and 10th quantiles of PNSDs, respectively.

The content was added in Line 235-236 in the manuscript:

"The PNSDs of ambient aerosols during the daytime and nighttime were shown in Figure S1."

Figure S1 and corresponding content were added in Line 30-36 in the appendix:

"The particle number size distributions (PNSDs) of ambient aerosols during the daytime and nighttime were shown in Figure S1. The particle number concentrations (PNCs) were $(1.99 \pm 1.28) \times 10^4$ and $(1.71 \pm 0.71) \times 10^4$ #/cm$^3$ in the day and at night, respectively."

[Figure]

**Figure S 1. The average particle number size distributions (PNSDs) measured by TDMPS (the black markers) during the (a) daytime (5:00-19:00) and (b) nighttime (20:00-4:00). The upper and lower edges of the grey area represent the 90th and 10th quantiles of the PNSDs, respectively.**

*2. Fig.2: The figure would profit from including the range of the deposition fraction (percentiles or SD). I would appreciate a corresponding figure for children and elderly in the supplement.*

[Response]: (1) The calculation process of DFs is as follows:

Input physiological parameters into the MPPD model to obtain a series of DFs of particles in the range of 10 nm - 10 μm. The hourly wet particle diameters were calculated by multiplying the hourly HGF data and the corresponding dry particle diameters, and then average them to obtain the average wet particle diameters during the sampling period. Then, based on the function between DF and particle size derived from the MPPD model, the DFs corresponding to the average wet particle sizes are obtained. Therefore, the range of DFs cannot be calculated here, and the range of the wet particle diameters are shown as follows.

(2) The size-resolved DFs for children and the elderly groups were added in the appendix.

**[Revise]: (1) The content was added in Line 231-232 in the manuscript:**

**"The wet diameters of hydrophobic and hygroscopic particles in the HRT were shown in Table S2."**

**Table S2 was added in Line 21-22 in the appendix:**

**Table S2 Wet diameters of hydrophobic and hygroscopic particles in the HRT**

| Dry diameter (nm) | Wet diameter (mean ± SD, nm) | | | |
|---|---|---|---|---|
| | ET | | TB/P | |
| | Hydrophobic | Hygroscopic | Hydrophobic | Hygroscopic |
| 12.6 | 12.7 ± 0.0 | 13.4 ± 0.4 | 13.1 ± 0.2 | 17.0 ± 2.1 |
| 14.2 | 14.3 ± 0.0 | 15.2 ± 0.5 | 14.9 ± 0.3 | 19.8 ± 2.6 |
| 16.1 | 16.2 ± 0.1 | 17.2 ± 0.5 | 17.0 ± 0.3 | 23.3 ± 3.2 |
| 18.2 | 18.3 ± 0.1 | 19.5 ± 0.6 | 19.3 ± 0.4 | 27.4 ± 3.7 |
| 20.5 | 20.7 ± 0.1 | 22.1 ± 0.6 | 22.0 ± 0.5 | 32.4 ± 4.3 |
| 23.2 | 23.4 ± 0.1 | 25.1 ± 0.6 | 25.1 ± 0.6 | 38.5 ± 4.7 |
| 26.2 | 26.5 ± 0.1 | 28.5 ± 0.6 | 28.7 ± 0.8 | 45.9 ± 5.0 |
| 29.7 | 30.0 ± 0.1 | 32.4 ± 0.7 | 32.8 ± 1.0 | 54.9 ± 5.4 |
| 33.6 | 33.9 ± 0.1 | 36.7 ± 0.7 | 37.5 ± 1.3 | 65.5 ± 5.9 |
| 38.0 | 38.4 ± 0.1 | 41.7 ± 0.8 | 42.9 ± 1.6 | 77.9 ± 6.8 |
| 42.9 | 43.4 ± 0.1 | 47.2 ± 0.8 | 49.1 ± 1.9 | 92.4 ± 7.9 |
| 48.5 | 49.0 ± 0.2 | 53.5 ± 1.0 | 56.2 ± 2.4 | 109.4 ± 9.3 |
| 54.9 | 55.4 ± 0.2 | 60.6 ± 1.1 | 64.3 ± 2.9 | 129.1 ± 11.0 |
| 62.0 | 62.7 ± 0.2 | 68.7 ± 1.2 | 73.5 ± 3.5 | 152.0 ± 12.8 |

| | | | | |
|---|---|---|---|---|
| 70.2 | 70.9 ± 0.2 | 77.8 ± 1.4 | 83.8 ± 4.0 | 178.8 ± 14.8 |
| 79.3 | 80.1 ± 0.2 | 88.1 ± 1.6 | 95.2 ± 4.7 | 209.8 ± 17.1 |
| 89.7 | 90.5 ± 0.2 | 99.7 ± 1.8 | 107.7 ± 5.3 | 245.8 ± 19.8 |
| 101.4 | 102.2 ± 0.2 | 113. 0 ± 2.1 | 121.4 ± 6.0 | 287.4 ± 22.9 |
| 114.7 | 115.5 ± 0.2 | 127.9 ± 2.4 | 136.3 ± 6.8 | 335.5 ± 26.7 |
| 129.7 | 130.4 ± 0.3 | 144.9 ± 2.8 | 152.6 ± 8.0 | 390.8 ± 31.5 |
| 146.7 | 147.4 ± 0.3 | 164.2 ± 3.3 | 170.8 ± 9.6 | 454.6 ± 37.5 |
| 165.8 | 166.5 ± 0.3 | 186.0 ± 4.0 | 191.1 ± 11.7 | 527.1 ± 44.8 |
| 187.5 | 188.2 ± 0.4 | 210.6 ± 4.8 | 214.1 ± 14.3 | 610.2 ± 53.9 |
| 212.0 | 212.6 ± 0.4 | 238.6 ± 5.8 | 240.4 ± 17.4 | 704.6 ± 64.9 |
| 239.8 | 240.4 ± 0.5 | 270.3 ± 6.9 | 270.6 ± 21.2 | 812.4 ± 77.7 |
| 271.2 | 271.8 ± 0.5 | 306.1 ± 8.3 | 305.0 ± 25.6 | 934.3 ± 92.3 |
| 306.6 | 307.2 ± 0.6 | 346.5 ± 9.7 | 344.3 ± 30.8 | 1072.0 ± 108.9 |
| 346.7 | 347.5 ± 0.7 | 392.3 ± 11.4 | 389.3 ± 37.0 | 1228.6 ± 127.5 |
| 392.0 | 392.8 ± 0.8 | 443.9 ± 13.3 | 440.6 ± 44.1 | 1405.2 ± 148.3 |
| 443.2 | 444.1 ± 0.9 | 502.2 ± 15.4 | 499.0 ± 52.4 | 1604.7 ± 171.4 |
| 501.3 | 502.2 ± 1.1 | 568.3 ± 17.7 | 565.9 ± 61.9 | 1830.9 ± 197.0 |
| 566.8 | 567.9 ± 1.2 | 642.8 ± 20.2 | 642.0 ± 73.0 | 2086.0 ± 225.3 |
| 640.9 | 642.1 ± 1.4 | 727.1 ± 23.1 | 728.9 ± 85.8 | 2374.4 ± 256.9 |
| 724.7 | 726.0 ± 1.6 | 822.2 ± 26.3 | 828.1 ± 100.7 | 2699.7 ± 292.1 |
| 819.5 | 821.0 ± 1.8 | 929.8 ± 29.9 | 941.4 ± 118.0 | 3067.6 ± 331.2 |
| 926.6 | 928.4 ± 2.0 | 1051.3 ± 33.8 | 1070.9 ± 138.5 | 3481.9 ± 374.7 |

(2)  The content was added in Line 253-254 in the manuscript:

"The size-resolved DFs for the children and the elderly groups were shown in Figure S2-S3."

Figure S2-S3 were shown in Line 37-50 in the appendix:

[Figure]

**Figure S 2. Size-resolved (a) head, (b) TB, (c) P, and (d) total deposition fractions (DFs) of particles under dry conditions (i.e., without considering hygroscopicity), and hydrophobic and hygroscopic particles in humid environments (i.e., considering hygroscopicity) for the children group. The black, blue, and red dots represent dry, hydrophobic, and hygroscopic particles, respectively. In Figure S2(a), the black dots representing DFs under dry conditions is hidden behind the blue dots representing DFs of hydrophobic particles, because these two sets of DFs are close to each other.**

[Figure]

**Figure S 3. Size-resolved (a) head, (b) TB, (c) P, and (d) total deposition fractions (DFs) of particles under dry conditions (i.e., without considering hygroscopicity), and hydrophobic and hygroscopic particles in humid environments (i.e., considering hygroscopicity) for the elderly group. The black, blue, and red dots represent dry, hydrophobic, and hygroscopic particles, respectively. In Figure S3(a), the black dots representing DFs under dry conditions is hidden behind the blue dots representing DFs of hydrophobic particles, because these two sets of DFs are close to each other.**

*3.    Fig.3: Is it possible to include the red line (as indicator for the hygroscopic and hydrophobic fraction) in all columns?*

[Response]: Figure 3 was modified according to the reviewer's suggestion.

**[Revise]: Figure 3 was shown in Line 325-330 in the manuscript:**

[Figure]

**Figure 3. Regional and total deposition doses for (a) children, (b) adults, and (c) the elderly with/without considering particle hygroscopicity. The dark blue columns represent doses without considering hygroscopicity. The light blue columns represent doses considering hygroscopicity. The red lines on the column represent the division of doses of hygroscopic (above the red line) and hydrophobic particles (below the red line). Numbers above each column mean the corresponding particle doses with a unit of $10^{10}$ #/day.**

*4. Fig.4: Is for this figure the average PSD (Fig. 1) or the daily PSD (corresponding to Fig. 5) used? How does it behave with the particle fraction above 1 μm in the head region? Fig. 2 shows an increasing deposition fraction for larger particles and there is a significant fraction of particles above 1 μm after hygroscopic growth.*

[Response]: (1) The deposition dose was calculated based on the hourly PNC data, and the corresponding hourly deposition doses were averaged and displayed as Figure 4. The deposition rate in Figure 5 was calculated in the same way.

(2) The head DFs of particles in the size range of 10 nm - 10 μm were shown in Figure R1. As displayed in Figure R1, for particles above 1 μm, the DF in the head increases first and then decreases as particle diameter increasing. The maximum DF appeared at about 2 μm. The statement describing the regional and total DFs of larger particles was added as follows.

[Figure]

Figure R 1 The size-resolved DFs in the head for the adults group. The particle density was set as 1.0 g/cm$^3$.

**[Revise]: The content was added in Line 454-459 in the manuscript:**

**"Besides, the deposition pattern of particles with diameters larger than 1 μm was not discussed here due to the lack of the measurement of the PNSD and hygroscopicity of coarse particles. While, as an important part of the ambient particle mass, the coarse particles may also make a significant contribution to the particle deposition in the HRT (Figure S9). The relevant research which may find out the impact of particle hygroscopicity on the deposition mass dose of coarse particles ought to be carried out in the future."**

**Figure S9 and the related statement were added in Line 90-95 in the appendix:**

**"As shown in Figure S9, a peak appeared at Dp = 2 - 3 μm in the DF curves of the head and P regions, which resulted in a peak in the total DF curve. It implied that particles with larger diameters may also have a significant contribution to the particle deposition in the human respiratory tract."**

[Figure]

**Figure S 9. The size-resolved regional and total DFs for the adult group. The particle density was set as 1.0 g/cm³.**

*5.  Fig.5: NO, CO, BC and OH should be introduced with the corresponding measurement method in the methods section. Is it possible to show a fraction of the BC mass to the mass of hydrophobic aerosol (calculated from the PSD)? This will be helpful in understanding the displayed data. The figure and interpretation would benefit from including the median deposition rate without considering hygroscopic growth as done in the other figures. For interpretation also, the total particle number concentration should be added and included into the discussion.*

[Response]: (1) Thanks for the reviewer's advice. The measurement methods of NO, CO, BC and OH radical were added in the manuscript.

(2) The fractions of the BC mass to the mass of hydrophobic particles were calculated and shown in the appendix.

(3) Figure 5 was modified and Figure S6 of the deposition rates without considering

hygroscopicity was added according to the reviewer's suggestion.

[Revise]: (1) The measurement methods of NO, CO, BC and OH radical were added in Line 106-110 in the manuscript:

"Besides, NO and CO was monitored by a NOx chemiluminescence analyzer (42i-TLE, Thermo Scientific) and a Trace Level Carbon Monoxide Analyzer (48iQ, Thermo Scientific), respectively (Chen et al., 2020). The BC mass concentrations were measured by a Multi-Angle Absorption Photometer (MAAP Model 5012, Thermo, Inc.) (Wang et al., 2019). The OH radical was measured by a Laser-induced fluorescence (LIF) (Tan et al., 2020)."

(2) Figure S8 was added in Line 86-89 in the appendix:

[Figure]

Figure S 8 The relationship between the mass proportion of BC in the hydrophobic particles and the hygroscopicity parameter (κ). The blue dots represent the κ corresponding to the top 5% highest mass proportions of BC in the hydrophobic particles.

(3) The content of Figure S6 was added in Line 374-375 in the manuscript:

"By comparison, the diurnal variations of the deposition rates of hygroscopic and hydrophobic particles without considering hygroscopicity were displayed in Figure S6."

Figure 5 and the related statement were modified in Line 370-372, 378-384, and 399-400 in the manuscript:

"In order to link human daily activities to the particulate matter deposition, the diurnal variations of deposition rates of hygroscopic and hydrophobic particles considering hygroscopicity for the adults group averaged over the entire field campaign were investigated and displayed in Figure 5."

[Figure]

**Figure 5. The diurnal variations of deposition rates considering hygroscopicity and particle number concentrations (PNCs) of (a) hygroscopic and (b) hydrophobic particles for the adults group. The red lines represent the median deposition rate. The upper and lower edges of the grey area represent the 75th and 25th quantiles of deposition rates, respectively. The black lines represent the average PNC. (c) The diurnal variations of the average concentrations of NO, CO, BC, and OH radical during the sampling period. The blue, red, black, and green lines represent NO, CO, BC, and OH radical, respectively.**

**"It is indicated that the deposition rates of particles were mainly influenced by the PNC."**

**Figure S6 was added in Line 77-81 in the appendix:**

[Figure]

**Figure S 6. The diurnal variations of deposition rates without considering hygroscopicity and particle number concentrations (PNCs) of (a) hygroscopic and (b) hydrophobic particles for the adults group. The red lines represent the median deposition rate. The upper and lower edges of the grey area represent the**

**75th and 25th quantiles of deposition rates, respectively. The black lines represent the average PNC.**

*6. L40: This sentence reads as life expectancy declines, which is not the case (Haidong et al., 2019).*

[Response]: According to the reviewer's suggestion, we revised the statement.

**[Revise]: Line 40-42 in the manuscript:**

**"Toxicological and epidemiological studies showed that ambient particles can result in the declining life expectancy and rising premature mortality (Chen et al., 2013; Correia et al., 2013; Dockery et al., 1993; Pope and Dockery, 2013; Pope et al., 2009)."**

*7. L.45: A reference for this statement will be beneficial.*

[Response]: Thanks for the reviewer's suggestion. A reference was added.

**[Revise]: Line 42-45 in the manuscript:**

**"Compared with coarse particles, submicron particles (i.e., particles with diameter ≤ 1 μm) have smaller sizes and larger specific surface areas, which tend to carry more toxic and harmful components and reach deeper into the human respiratory tract (HRT) (Oberdorster, 2001)."**

*8. L.74: this statement is quite general. There are also many measurements of hygroscopicity with e.g. CCN counters.*

[Response]: Thanks for the reviewer's comment. The related statement was corrected.

**[Revise]: The content was modified in Line 73-76 in the manuscript:**

**"Hygroscopicity measurements of ambient aerosols under subsaturated conditions are mainly conducted by Humidity Tandem Differential Mobility Analyzer (H-TDMA) (Farkas et al., 2022; Kristensson et al., 2013; Londahl et al., 2009) or Differential Aerosol Sizing and Hygroscopicity Spectrometer Probe (DASH-SP) (Cavaleiro Rufo et al., 2016; Youn et al., 2016)."**

*9. L.161: Is deposition and clearance not the output parameter of the MPPD model?*

[Response]: Deposition/clearance is an input option in MPPD v3.04, which is used to select whether to perform clearance calculations in addition to deposition calculations in the human respiratory tract. Please refer to Line 196-197 in the manuscript: "It should be noted that any clearance mechanisms were not considered in this study, hence our results show the upper limit of exposure."

*10. L.165: The particle range in model is until 1 μm but PSD in Fig. 1(b) reaches up to 1.2 μm. What is the expected effect of neglecting the tail of the PSD?*

[Response]: We thank the reviewer for pointing out this typo. The particle range

selected in the MPPD model was 0.01-10.0 μm rather than 0.01-1.0 μm.

**[Revise]: The content was corrected in Line 182-183 in the manuscript:**

**"The particle diameter range was set as 0.01 - 10.0 μm."**

*11. L318 and Fig.2: What is the potential explanation for the underestimation of particles below 20 nm in P?*

[Response]: The DFs of hygroscopic and hydrophobic particles with considering hygroscopicity (hereinafter referred to as "wet particles") were obtained based on the DFs of particles without considering hygroscopicity (hereinafter referred to as "dry particles"). More specifically, it was considered that the DF of wet particles with $D_p$ after hygroscopic growth was equal to that of dry particles with the same diameter ($D_p$). Therefore, the DF curve of wet particles corresponds to shifting the particle size range of the DF curve of dry particles to the right. For clarity, the abscissa of Figure 2 refers to the dry diameter of particles. Therefore, the phenomenon of the underestimation of particles below 20 nm in the P region occurred.

**[Revise]: The statement was added in Line 208-213 and Line 231-232 in the manuscript:**

**"The DFs of hygroscopic and hydrophobic particles with considering hygroscopicity (hereinafter referred to as "wet particles") were obtained based on the DFs of particles without considering hygroscopicity (hereinafter referred to as "dry particles"). More specifically, it is considered that the DF of wet particles with Dp after hygroscopic growth was equal to that of dry particles with the same diameter (Dp). Therefore, the DF curve of wet particles corresponds to shifting the particle size range of the DF curve of dry particles to the right."**

**"The wet diameters of hydrophobic and hygroscopic particles in the HRT were shown in Table S2."**

*12. L.347: The conclusion from the presence of hygroscopic newly formed aerosol is somehow confusing, since the study points out that hygroscopic aerosol is less effective deposited in the HRT. The factor is the larger number concentration, not the aerosol hygroscopicity (as mentioned above including the total number concertation in Fig.5 will make this clearer). I also wonder why the aerosol disappeared in the morning hours (e.g. Fig.S1 2014/06/28 around 8 am)?*

[Response]: (1) Although the particle hygroscopicity had greater impact on the deposition of the hygroscopic particles than that of the hydrophobic particles, the hygroscopic particles contributed more to the particle deposition in the HRT due to the larger PNC (please refer to Figure 3 in the manuscript).

(2) Thanks for the reviewer's advice. Figure 5 and the related statement were modified as the reviewer's suggestion. Please refer to the revise (3) in the fifth specific comment.

(3) Aerosols disappeared due to the change of meteorological conditions, such as the increase of wind speeds.

**[Revise]: The content was modified in Line 392-395 in the manuscript:**

**"Our previous study showed that the NPF events and subsequent growth produced a large amount of hygroscopic and internally mixed particles (Wu et al., 2017b), thus increasing the PNC (as shown in the black line in Figure 5(a)) and leading to the enhanced deposition rate of hygroscopic particles in the day."**

*13. L.356: Is the low boundary layer height meant as a general feature of the PBL or is it somehow measured or modeled during the campaign? A reference here would be nice.*

[Response]: The obvious diurnal variation of the planetary boundary layer (PBL) height is a general feature of the PBL. The PBL is always lower at night than that during the daytime, which was not measured or modeled in this study. The reference was added according to the reviewer's advice.

**[Revise]: The content was modified in Line 406-408 in the manuscript:**

**"The strong primary emissions (such as the biomass burning process (Wu et al., 2017a), weak chemical processes, and low boundary layer height (Zhang et al., 2014) resulted in the increased hydrophobic particle number concentration."**

*14. L.363: This conclusion cannot be drawn that straight forward. What about the other aerosol during the peak concentrations of BC mass? How does the K value for BC compare to literature studies on the hygroscopicity of BC? An indication in Fig.S2 which are the "5% highest BC periods" chosen and also the BC fraction compared to all aerosol mass will be beneficial.*

[Response]: (1) Thanks for the reviewer's suggestion. In heavy polluted events, ambient aerosols with high BC mass concentration may also consist of high concentration level of secondary species. Therefore, the related statement was modified in the manuscript.

(2) Pure BC particles is always regarded as completely hydrophobic, so the κ of BC is set as 0 (Riemer et al., 2010).

(3) Based on the reviewer's suggestion, we calculated the mass proportion of BC in the hydrophobic particles to replace the BC mass concentrations, and corresponding κ was modified. Besides, the former Figure S2 was deleted from the appendix.

**[Revise]: The related statement was modified in Line 415-420 in the manuscript:**

**"Since BC primarily comes from combustion processes, the κ corresponding to the top 5% highest mass proportion of BC in the hydrophobic particles was employed to represent the κ value of primary emission particles (Figure S8). The κ of BC with Dp = 50 nm was 0.20 ± 0.12, which was lower than that of the similar size particles during the NPF events (κ = 0.37 ± 0.03) in our previous**

**publications (Wu et al., 2017b). In this study, the total DF for adults was ~ 0.258 when κ = 0.20, while it dropped to ~ 0.217 when κ = 0.37, with a decline change of 15.9%."**

**The Figure S8 was added in Line 86-89 in the appendix:**

[Figure]

**Figure S 8. The relationship between the mass proportion of BC in the hydrophobic particles and the hygroscopicity parameter (κ). The blue dots represent the κ corresponding to the top 5% highest mass proportions of BC in the hydrophobic particles.**

*15. L.367: I do not see the benefit of the discussion; the PSD and PNC are not the same for freshly emitted and aged aerosol. Further, hydrophobic and hygroscopic particles can be emitted from the same source.*

[Response]: Thanks for the reviewer's advice. The related statement was deleted in the manuscript.

*16. L.388: The discussion on the atmospheric oxidation capacity could be more addressed in the results section to strengthen this conclusion.*

[Response]: Thanks for the reviewer's advice. Some discussion was added in the manuscript.

**[Revise]: The content was added in Line 392-395 in the manuscript:**

**"For instance, a previous study pointed out that the atmospheric oxidation capacity may enhance the particle formation by oxidizing the trace gases emitted from sources (such as biomass burning) and producing low-volatility condensable vapors, which nucleates in the plume (Wu et al., 2017a) Therefore, the ultrafine particles may be produced during transport (Wu et al., 2017a)."**

*17. L.390: What are the strong primary emission sources over night?*

[Response]: According to our previous study (Wu et al., 2017a), the strong primary emission source is biomass burning.

**[Revise]: The content was modified in Line 406-408 in the manuscript:**

**"The strong primary emissions (such as the biomass burning process (Wu et al., 2017a)), weak chemical processes, and low boundary layer height (Zhang et al., 2014) resulted in the increased hydrophobic particle number concentration."**

*18. L.392: This study does not provide insights about aging state, it shows the difference between hydrophobic and hygroscopic particles. Thus, I recommend to rephrase the sentence from "aged aerosol" to hygroscopic.*

[Response]: Thanks for the reviewer's advice. The statement was modified in the manuscript.

**[Revise]: The content was modified in Line 440-442 in the manuscript:**

**"Additionally, fresh emitted particles have lower hygroscopicity and higher DFs compared with hygroscopic particles with the same diameter and concentration, which may result in higher deposition."**

*19. L.396: I do not see this aspect covered by the manuscript. Maybe a rephrasing closer to the actual research performed will help.*

[Response]: Thanks for the reviewer's advice. The statement was deleted in the manuscript.

*20. L.399: Data availability – maybe the authors want to consider a publication of the data on an open accessible research data repository after publication.*

[Response]: Thanks for the reviewer's suggestion. We will submit the data to the open accessible research data repository after the publication of our manuscript.

**[Revise]: Line 461-462 in the manuscript:**

**"*Data availability.* The data presented in this article can be accessed on the open accessible research data repository."**

*Technical comments:*

*1. There are some citations doubled in single sentences. Some of the cited literature seems to be not the primary literature.*

[Response]: Thanks for the reviewer's correction. The references were checked and modified.

*2. L.140: Consider to reconstruct the sentence.*

[Response]: Thanks for the reviewer's advice. This sentence was reconstructed as follows.

**[Revise]: Line 159-162 in the manuscript:**

**"The hydrophobic particles in urban environments have been interpreted as**

originating from freshly emitted soot and vehicle exhaust, while the hygroscopic particles have been regarded to experience long-distance transport (Baltensperger, 2002; Swietlicki et al., 2008)."

**Reference**

Baltensperger, U.: Urban and rural aerosol characterization of summer smog events during the PIPAPO field campaign in Milan, Italy, Journal of Geophysical Research, 107, 10.1029/2001jd001292, 2002.

Bennett, W. D., Zeman, K. L., and Jarabek, A. M.: Nasal contribution to breathing and fine particle deposition in children versus adults, J Toxicol Environ Health A, 71, 227-237, 10.1080/15287390701598200, 2008.

Cavaleiro Rufo, J., Madureira, J., Paciencia, I., Slezakova, K., Pereira Mdo, C., Aguiar, L., Teixeira, J. P., Moreira, A., and Oliveira Fernandes, E.: Children exposure to indoor ultrafine particles in urban and rural school environments, Environ Sci Pollut Res Int, 23, 13877-13885, 10.1007/s11356-016-6555-y, 2016.

Chen, S., Wang, H., Lu, K., Zeng, L., Hu, M., and Zhang, Y.: The trend of surface ozone in Beijing from 2013 to 2019: Indications of the persisting strong atmospheric oxidation capacity, Atmospheric Environment, 242, 10.1016/j.atmosenv.2020.117801, 2020.

Chen, Y., Ebenstein, A., Greenstone, M., and Li, H.: Evidence on the impact of sustained exposure to air pollution on life expectancy from China's Huai River policy, Proc Natl Acad Sci U S A, 110, 12936-12941, 10.1073/pnas.1300018110, 2013.

Ching, J. and Kajino, M.: Aerosol mixing state matters for particles deposition in human respiratory system, Sci Rep, 8, 8864, 10.1038/s41598-018-27156-z, 2018.

Correia, A. W., Pope, C. A., 3rd, Dockery, D. W., Wang, Y., Ezzati, M., and Dominici, F.: Effect of air pollution control on life expectancy in the United States: an analysis of 545 U.S. counties for the period from 2000 to 2007, Epidemiology, 24, 23-31, 10.1097/EDE.0b013e3182770237, 2013.

Daigle, C. C., Chalupa, D. C., Gibb, F. R., Morrow, P. E., Oberdorster, G., Utell, M. J., and Frampton, M. W.: Ultrafine particle deposition in humans during rest and exercise, Inhal Toxicol, 15, 539-552, 10.1080/08958370304468, 2003.

Dockery, D. W., Pope, C. A., 3rd, Xu, X., Spengler, J. D., Ware, J. H., Fay, M. E., Ferris, B. G., Jr., and Speizer, F. E.: An association between air pollution and mortality in six U.S. cities, The New England journal of medicine, 329, 1753-1759, 10.1056/nejm199312093292401, 1993.

Farkas, A., Furi, P., Then, W., and Salma, I.: Effects of hygroscopic growth of ambient urban aerosol particles on their modelled regional and local deposition in healthy and COPD-compromised human respiratory system, Science of the Total Environment, 806, 10.1016/j.scitotenv.2021.151202, 2022.

Hu, M., Peng, J., Sun, K., Yue, D., Guo, S., Wiedensohler, A., and Wu, Z.: Estimation of size-resolved ambient particle density based on the measurement of aerosol number, mass, and chemical size distributions in the winter in Beijing, Environ

Sci Technol, 46, 9941-9947, 10.1021/es204073t, 2012.

Khlystov, A., Stanier, C., and Pandis, S. N.: An Algorithm for Combining Electrical Mobility and Aerodynamic Size Distributions Data when Measuring Ambient Aerosol Special Issue ofAerosol Science and Technologyon Findings from the Fine Particulate Matter Supersites Program, Aerosol Science and Technology, 38, 229-238, 10.1080/02786820390229543, 2004.

Kristensson, A., Rissler, J., Londahl, J., Johansson, C., and Swietlicki, E.: Size-Resolved Respiratory Tract Deposition of Sub-Micrometer Aerosol Particles in a Residential Area with Wintertime Wood Combustion, Aerosol and Air Quality Research, 13, 24-35, 10.4209/aaqr.2012.07.0194, 2013.

Londahl, J., Massling, A., Pagels, J., Swietlicki, E., Vaclavik, E., and Loft, S.: Size-resolved respiratory-tract deposition of fine and ultrafine hydrophobic and hygroscopic aerosol particles during rest and exercise, Inhalation Toxicology, 19, 109-116, 10.1080/08958370601051677, 2007.

Londahl, J., Massling, A., Swietlicki, E., Brauner, E. V., Ketzel, M., Pagels, J., and Loft, S.: Experimentally Determined Human Respiratory Tract Deposition of Airborne Particles at a Busy Street, Environmental Science & Technology, 43, 4659-4664, 10.1021/es803029b, 2009.

MPPD: Multiple-Path Particle Dosimetry Model: https://www.ara.com/mppd/, last access: 2022.

Oberdorster, G.: Pulmonary effects of inhaled ultrafine particles, International Archives of Occupational and Environmental Health, 74, 1-8, 2001.

Pope, C. A., 3rd and Dockery, D. W.: Air pollution and life expectancy in China and beyond, Proc Natl Acad Sci U S A, 110, 12861-12862, 10.1073/pnas.1310925110, 2013.

Pope, C. A., III, Ezzati, M., and Dockery, D. W.: Fine-Particulate Air Pollution and Life Expectancy in the United States, NEW ENGLAND JOURNAL OF MEDICINE, 360, 376-386, 10.1056/NEJMsa0805646, 2009.

Riemer, N., West, M., Zaveri, R., and Easter, R.: Estimating black carbon aging time-scales with a particle-resolved aerosol model, Journal of Aerosol Science, 41, 143-158, 10.1016/j.jaerosci.2009.08.009, 2010.

Salma, I., Füri, P., Németh, Z., Balásházy, I., Hofmann, W., and Farkas, Á.: Lung burden and deposition distribution of inhaled atmospheric urban ultrafine particles as the first step in their health risk assessment, Atmospheric Environment, 104, 39-49, 10.1016/j.atmosenv.2014.12.060, 2015.

Swietlicki, E., Hansson, H. C., Hameri, K., Svenningsson, B., Massling, A., McFiggans, G., McMurry, P. H., Petaja, T., Tunved, P., Gysel, M., Topping, D., Weingartner, E., Baltensperger, U., Rissler, J., Wiedensohler, A., and Kulmala, M.: Hygroscopic properties of submicrometer atmospheric aerosol particles

measured with H-TDMA instruments in various environments - a review, Tellus Series B-Chemical and Physical Meteorology, 60, 432-469, 10.1111/j.1600-0889.2008.00350.x, 2008.

Tan, Z., Hofzumahaus, A., Lu, K., Brown, S. S., Holland, F., Huey, L. G., Kiendler-Scharr, A., Li, X., Liu, X., Ma, N., Min, K. E., Rohrer, F., Shao, M., Wahner, A., Wang, Y., Wiedensohler, A., Wu, Y., Wu, Z., Zeng, L., Zhang, Y., and Fuchs, H.: No Evidence for a Significant Impact of Heterogeneous Chemistry on Radical Concentrations in the North China Plain in Summer 2014, Environ Sci Technol, 54, 5973-5979, 10.1021/acs.est.0c00525, 2020.

Wang, T., Du, Z., Tan, T., Xu, N., Hu, M., Hu, J., and Guo, S.: Measurement of aerosol optical properties and their potential source origin in urban Beijing from 2013-2017, Atmospheric Environment, 206, 293-302, 10.1016/j.atmosenv.2019.02.049, 2019.

Wu, Z., Zheng, J., Wang, Y., Shang, D., Du, Z., Zhang, Y., and Hu, M.: Chemical and physical properties of biomass burning aerosols and their CCN activity: A case study in Beijing, China, Sci Total Environ, 579, 1260-1268, 10.1016/j.scitotenv.2016.11.112, 2017a.

Wu, Z. J., Ma, N., Größ, J., Kecorius, S., Lu, K. D., Shang, D. J., Wang, Y., Wu, Y. S., Zeng, L. M., Hu, M., Wiedensohler, A., and Zhang, Y. H.: Thermodynamic properties of nanoparticles during new particle formation events in the atmosphere of North China Plain, Atmospheric Research, 188, 55-63, 10.1016/j.atmosres.2017.01.007, 2017b.

Youn, J. S., Csavina, J., Rine, K. P., Shingler, T., Taylor, M. P., Saez, A. E., Betterton, E. A., and Sorooshian, A.: Hygroscopic Properties and Respiratory System Deposition Behavior of Particulate Matter Emitted By Mining and Smelting Operations, Environmental Science & Technology, 50, 11706-11713, 10.1021/acs.est.6b03621, 2016.

Zhang, Y., Zhang, S., Huang, C., Huang, K., Gong, Y., and Gan, Q.: Diurnal variations of the planetary boundary layer height estimated from intensive radiosonde observations over Yichang, China, Science China Technological Sciences, 57, 2172-2176, 10.1007/s11431-014-5639-5, 2014.

---

## Author Comment (AC2)

**Response to reviewers**

Thanks for reviewers' careful reading and constructive comments and suggestions. We made every effort to respond to reviewers' questions point to point, and revised our manuscript and appendix according to their comments. For clarity, reviewers' comments are shown in *black italic font*. The response is shown in blue normal font. The modified content in the manuscript and/or the appendix is shown in **green bold font**.

**Anonymous Referee #2**

Referee comment on "Impact of water uptake and mixing state on submicron particles deposition in the human respiratory tract (HRT): Based on explicit hygroscopicity measurements at HRT-like conditions" by Ruiqi Man et al., EGUsphere, https://doi.org/10.5194/egusphere-2022-256-RC2, 2022

*Summary:*

*In this work the authors describe measurement of the hygroscopic growth of externally mixed particles from the North China Plain. They use these data in conjunction with a lung deposition model to predict the effect of hygroscopic growth on deposition in the respiratory tract. The results show that in total, dose was reduced when hygroscopic growth effects were considered as the more numerous smaller particles, that deposit via diffusion mechanisms, deposited less effectively. Variations were seen across the size range, with smaller particles showing a reduced likelihood to deposit, while larger particles were more likely to deposit.*

*Overall, this paper goes some way towards showing the importance of considering hygroscopic growth, but the extent of new insights is limited. The effects are reported to be rather small so an improved sensitivity analysis and consideration of uncertainties is needed to validate and support the conclusions. Some specific points towards this are detailed below:*

*1. Deposition fraction is on a particle number basis, and the conclusions connect the dose with the number of particles. The authors should consider reporting dose on a mass deposition basis, which will significantly increase the contributions of the larger particles on deposited dose.*

[Response]: Thanks for the reviewer's suggestion. The particle deposition can be weighed by particle number concentrations, mass concentrations, and/or surface area concentrations. Evaluating the deposition dose on a mass basis is definitely an important task in exploring the health risk of particles (such as in toxicological studies). However, due to the lack of the measurement of the PNSD and hygroscopicity of coarse particles, the deposition mass dose of larger particles cannot

be calculated in this study. We used the particle number dose as a weight for the reason that the measurement object was submicron particles rather than coarse particles. As the predominant particle type by number in ambient submicron particles, ultrafine particles contribute insignificantly to mass (Xia et al., 2009), but they do great harm to human health (Elsaesser and Howard, 2012; Englert, 2004; Oberdorster, 2001; Sioutas et al., 2005). Therefore, the deposition number dose can highlight the health risk of ultrafine particles. As mentioned above, we used the deposition number dose of particulate matters as a measure to study the particle deposition. The relevant statement was added in the manuscript.

**[Revise]: The statement was added in Line 454-459 in the manuscript:**

**"Besides, the deposition pattern of particles with diameters larger than 1 µm was not discussed here due to the lack of the measurement of the PNSD and hygroscopicity of coarse particles. While, as an important part of the ambient particle mass, coarse particles may also make a significant contribution to the particle deposition in the HRT (Figure S9). The related research to find out the impact of the particle hygroscopicity on the deposition mass dose of coarse particles ought to be carried out in the future."**

**Figure S9 and related content were added in Line 90-95 in the appendix:**

**"As shown in Figure S9, a peak appeared at $D_p$ = 2 - 3 µm in the DF curves of the head and P regions, which resulted in a peak in the total DF curve. It implied that particles with larger diameters may also have a significant contribution to the particle deposition in the human respiratory tract.**

[Figure]

**Figure S 9. The size-resolved regional and total DFs for the adults group. The particle density was set as 1.0 g/cm³."**

*2. Does the lung deposition model change the density of the particles as they grow due to water uptake? A density of 1.5 g/cm3 is high for hygroscopic particles at >90% RH. I suggest a sensitivity analysis be performed to compare the difference in deposition for 1.0 and 1.5 g/cm3 particle distributions.*

[Response]: (1) Thanks for the reviewer's comment. The change of the particle density during the water uptake was not included in the MPPD model.

(2) Due to the lack of the particle density measurement during the sampling period, we compared the differences between the size-resolved DFs of particles with $\rho_p$ = 1.0 g/cm$^3$ vs. $\rho_p$ = 1.5 g/cm$^3$ for adults. The results and discussions of the sensitivity analysis of the particle density were added as follows.

**[Revise]: The statement was added in Line 291-293 in the manuscript:**

**"It should be noted that the particle density would change during hygroscopic growth, which was not considered in the calculation due to the lack of the measurement of the particle density. The sensitivity analysis of the particle density on the regional DFs was shown in Figure S4."**

**Figure S4 and related statement were added in Line 51-68 in the appendix:**

**"The particle density mainly affects the probability of inertial impaction during the particle deposition process, which can be evaluated by using the dimensionless Stokes number (Stk), defined as Eq (S1) (Pramod et al., 2011):**

$$\text{Stk} = \frac{\rho_p d_p{}^2 C_c U}{18 \eta d_f} \tag{S1}$$

**where $\rho_p$ is the density of the particle. The Stokes number is the basic parameter describing the inertial impaction mechanism. A larger Stokes number implies a higher probability of deposition by impaction (Pramod et al., 2011).**

**Due to the lack of the density measurement of particles during the sampling period, the differences between the size-resolved DFs of particles with $\rho_p$ = 1.0 g/cm$^3$ vs. $\rho_p$ = 1.5 g/cm$^3$ for adults were compared. As displayed in Figure S4, the particle density has great influence on the particle deposition in the head and P regions for larger submicron particles. The average DF differences in the head, TB, P, and the whole HRT were (11.1 ± 13.9)%, (0.5 ± 0.8)%, (3.8 ± 6.4)%, and (4.2 ± 6.5)%, respectively. Therefore, the measurement or estimation of the particle density during the particle hygroscopic growth is of great importance in calculating the particle deposition in human bodies.**

[Figure]

**Figure S 4. Size-resolved regional deposition fractions (DFs) of particles with density (ρ) = 1.0 vs. 1.5 g/cm$^3$ for the adults group. The blue, green, and purple markers represent the DFs of particles with ρ = 1.0 g/cm$^3$ in the head, TB, and P, respectively. The blue, green, and purple lines represent the DFs of particles with ρ = 1.5 g/cm$^3$ in the head, TB, and P, respectively."**

**The statement was added in Line 452-454 in the manuscript:**

**"Due to the limited measurements and physiological parameters, some vital factors which may have effect on the particle deposition in the HRT were not considered in this study, such as gender, the exercise level, and the particle density."**

*3. How was the dry size of the particles determined in the hygroscopic growth measurements? Were any shape correction factors considered?*

[Response]: (1) As seen in Figure R1, the H-TDMA consists of two differential mobility analyzers (DMAs) and two condensation particle counters (CPCs). The monodisperse aerosol sizes cut with mobility diameters (30, 50, 100, 150, 200, and 250 nm in this study) were selected in turn by the first DMA under dry conditions (RH < 10%). Then, the aerosols passed through a humidifier with a controlled higher RH, and the size distributions over wet mobility diameters were measured with the second DMA (Duplissy et al., 2011).

[Figure]

Figure R 1. Set-up of the H-TDMA (Hennig et al., 2005)

(2) No shape correction factor was considered in this study. The ambient aerosols at this sampling site mainly consisted of secondary aerosols (Wu et al., 2017), which are mostly spherical. Irregular particles, such as sea salt and dust, contributed little to aerosols collected at the sampling site. In addition, irregular particles generally exist in coarse mode particles rather than submicron particles. Therefore, The shape factor $\mathcal{X}$ was set as 1.

**[Revise]: The content was added in Line 148-155 in the manuscript:**

**"To match the particle size range in the MPPD model, the electrical mobility diameter was converted to aerodynamic diameter by Eq (4) (Khlystov et al., 2004):**

$$\mathbf{d_a} = \mathbf{d_m} \sqrt{\mathcal{X} \times \frac{\rho \times C_{c(dm)}}{C_{c(da)}}} \qquad (4)$$

**where $\mathbf{d_a}$ and $\mathbf{d_m}$ is the particle aerodynamic diameter (nm) and electrical mobility diameter (nm), respectively. $\rho$ is the particle density (1.5 g cm⁻³ in this study (Hu et al., 2012)). $\mathcal{X}$ is the shape factor. $C_c$ is the Cunningham slip correction factor for a certain diameter. Similar to other studies, the shape factor $\mathcal{X}$ is assumed as 1 and $C_c$ is neglected in the calculation (Khlystov et al., 2004; Hu et al., 2012). Therefore, the electrical mobility diameter (in the range of 10.3 - 756.6 nm) was converted to the aerodynamic diameter (in the range of 12.6 - 926.6 nm)."**

*4. How accurate is the RH measured in the HTDMA? How stable is the RH? At the high RH of these measurements, even fractions of a % of RH can lead to significant changes in the size of the particles and will introduce uncertainty in the results.*

[Response]: The accuracy and stability of the HH-TDMA were studied by Hennig et al (2005). The RH in the second DMA reached an absolute accuracy of ±1.2% for 98% and a long-term stability of ± 0.1-0.4% of set point values (Hennig et al., 2005).

**[Revise]: The content was added in Line 114-116 in the manuscript:**

**"The RH in the second DMA reached an absolute accuracy of ±1.2% for 98% and a long-term stability of ± 0.1-0.4% of set point values (Hennig et al., 2005)."**

*5. On line 103, HH-TDMA is referred to – what does the second "H" stand for?*

[Response]: The HH-TDMA is the abbreviation of the high humidity tandem differential mobility analyzer (please refer to Line 91-92 in the manuscript). Therefore, the second 'H' stands for 'humidity'.

*6. Line 84 – a constant value of kappa with RH does not indicate an ideal solution. It indicates that the effective molar volume of the solute does not vary with RH.*

[Response]: Thanks for the reviewer's correction. The related expression was modified according to the reviewer's advice.

**[Revise]: Line 83-84 in the manuscript:**

**"It was further assumed that κ was independent of RH on the premise that the effective molar volume of the solute does not vary with RH."**

**Reference**

Duplissy, J., DeCarlo, P. F., Dommen, J., Alfarra, M. R., Metzger, A., Barmpadimos, I., Prevot, A. S. H., Weingartner, E., Tritscher, T., Gysel, M., Aiken, A. C., Jimenez, J. L., Canagaratna, M. R., Worsnop, D. R., Collins, D. R., Tomlinson, J., and Baltensperger, U.: Relating hygroscopicity and composition of organic aerosol particulate matter, Atmospheric Chemistry and Physics, 11, 1155-1165, 10.5194/acp-11-1155-2011, 2011.

Elsaesser, A. and Howard, C. V.: Toxicology of nanoparticles, Advanced Drug Delivery Reviews, 64, 129-137, 10.1016/j.addr.2011.09.001, 2012.

Englert, N.: Fine particles and human health - a review of epidemiological studies, Toxicology Letters, 149, 235-242, 10.1016/j.toxlet.2003.12.035, 2004.

Hennig, T., Massling, A., Brechtel, F. J., and Wiedensohler, A.: A tandem DMA for highly temperature-stabilized hygroscopic particle growth measurements between 90% and 98% relative humidity, Journal of Aerosol Science, 36, 1210-1223, 10.1016/j.jaerosci.2005.01.005, 2005.

Hu, M., Peng, J., Sun, K., Yue, D., Guo, S., Wiedensohler, A., and Wu, Z.: Estimation of size-resolved ambient particle density based on the measurement of aerosol number, mass, and chemical size distributions in the winter in Beijing, Environ Sci Technol, 46, 9941-9947, 10.1021/es204073t, 2012.

Khlystov, A., Stanier, C., and Pandis, S. N.: An Algorithm for Combining Electrical Mobility and Aerodynamic Size Distributions Data when Measuring Ambient Aerosol Special Issue ofAerosol Science and Technologyon Findings from the Fine Particulate Matter Supersites Program, Aerosol Science and Technology, 38, 229-238, 10.1080/02786820390229543, 2004.

Oberdorster, G.: Pulmonary effects of inhaled ultrafine particles, International Archives of Occupational and Environmental Health, 74, 1-8, 2001.

Pramod, K., Paul B., and Klaus W.: Aerosol measurement: principles, techniques, and applications (Third Edition), John Wiley & Sons, Inc, Hoboken, the United States, 2011.

Sioutas, C., Delfino, R. J., and Singh, M.: Exposure assessment for atmospheric ultrafine particles (UFPs) and implications in epidemiologic research, Environmental Health Perspectives, 113, 947-955, 10.1289/ehp.7939, 2005.

Wu, Z. J., Ma, N., Größ, J., Kecorius, S., Lu, K. D., Shang, D. J., Wang, Y., Wu, Y. S., Zeng, L. M., Hu, M., Wiedensohler, A., and Zhang, Y. H.: Thermodynamic properties of nanoparticles during new particle formation events in the atmosphere of North China Plain, Atmospheric Research, 188, 55-63, 10.1016/j.atmosres.2017.01.007, 2017.

Xia, T., Li, N., and Nel, A. E.: Potential Health Impact of Nanoparticles, Annual Review of Public Health, 30, 137-150,

10.1146/annurev.publhealth.031308.100155, 2009.

---

## Author Response (AR2)

**Response to reviewers**

Thanks for reviewers' helpful comments and suggestions again. We made every effort to respond to reviewers' questions point to point, and revised our manuscript and appendix according to their comments. For clarity, reviewers' comments are shown in *black italic font*. The response is shown in blue normal font. The modified content in the manuscript and/or the appendix is shown in **green bold font**.

*Anonymous Referee #1*

General comment:

*The authors replied most of the reviewer's questions sufficient and the manuscript is greatly improved, however, there is in Review#1 and Review#2 the open question of the novelty of the study. I.e., in Review#1 "The study will profit from a more detailed comparison to previous studies, both on ambient aerosol properties and the HRT deposition. This comparison will also allow the authors to clearly define the novelty of their study" and in Review#2 "the extent of new insights is limited". This point is not addressed in the reply and revised manuscript. I recommend the authors to consider these open questions raised by the two reviewers.*

[Response]: Thanks for the reviewers' suggestion. To better understand the novelty of this study, a literature survey about ambient particle hygroscopic properties and the particle deposition in the HRT was conducted.

(1) As for the particle hygroscopicity, the result measured by HH-TDMA at RH = 98% in this study was similar to those measured by H-TDMA or HH-TDMA at RH = 87% - 98.5%. Detailed description can be found in the appendix (see Line 19-29). Although the difference of the particle hygroscopic property determined between RH = 90% and RH = 98% cannot be evaluated due to the lack of simultaneous measurements, previous studies have found that the presence of surface active, slightly soluble substances, and the co-condensation of semi-volatile soluble organic compounds can result in the humidity-dependent characteristic of $\kappa$ (Topping and Mcfiggans, 2012; Wex et al., 2009; Wu et al., 2013). Therefore, the result measured by HH-TDMA at RH = 98% ought to be closer to the hygroscopicity in the HRT-conditions.

(2) Up to now, the published research which assessed the particle deposition in the HRT with considering hygroscopicity are mostly based on the assumed values of the hygroscopic parameter (Voliotis and Samara, 2018) or estimations by parametric methods (Ching and Kajino, 2018; Hussein et al., 2013; Mitsakou et al., 2007; Haddrell et al., 2015; Vu et al., 2018). By comparison, the direct particle hygroscopicity measurements can capture the real and high-time-resolution features of ambient particles' hygroscopic growth in the HRT, and reveal the diurnal variations of the particulate matter deposition in human bodies.

(3) To our best knowledge, there are only limited studies exploring the impact of the hygroscopic growth of ambient particles on the particle deposition by direct hygroscopicity measurements. Moreover, hygroscopicity measurements using the H-TDMA (Londahl et al., 2009; Farkas et al., 2022; Vu et al., 2015; Kristensson et al., 2013)

or DASH-SP (Youn et al., 2016) in these studies were all conducted at relatively lower RH (~ 90%) compared to that in the HRT (RH = 99.5%).

(4) Besides, the field campaign was carried out in the North China Plain, a polluted area with high population density and strong primary emissions. The particle deposition estimation with considering hygroscopicity in this area has not been reported yet.

In conclusion, this study first combined the explicit hygroscopicity measurements at HRT-like conditions by HH-TDMA with the MPPD model, which enriches the understanding of the influence of water uptake and hygroscopic growth on submicron particles deposition in the HRT.

**[Revise]: Line 19-29 in the appendix:**

**"The particle hygroscopicity parameter (Kappa, κ) in this study and previous studies measured in rural sites in the North China Plain (NCP) was shown in Table S1. The average size-resolved κ was in the range of 0.24 - 0.32 during the sampling period. The hygroscopic properties of particles in this study was similar to those determined in the NCP in summer, such as in Wuqing (Liu et al., 2011) and Xianghe (Zhang et al., 2016), which was higher than that measured in winter, such as in Dingxing (Shi et al., 2022). It can be explained that the mass fraction of organic matters with relatively weak hygroscopicity was higher in winter, while secondary inorganic aerosols with strong hygroscopicity made higher contribution in summer (Sun et al., 2015). Besides, the particle hygroscopicity increased as the particle diameter increasing, which was in accordance with previous studies measured in urban and rural sites (Swietlicki et al., 2008)."**

**Line 61-85 in the manuscript:**

**"To date, many studies have assessed the effects of hygroscopicity on ambient particle deposition in the HRT based on assumed values of the hygroscopic parameter (Kappa, κ) representing non-hygroscopic, nearly hydrophobic, and hygroscopic particles (Voliotis and Samara, 2018) or estimations by parametric methods (Ching and Kajino, 2018; Hussein et al., 2013; Mitsakou et al., 2007; Haddrell et al., 2015; Vu et al., 2018) . However, it is well-known that continental aerosols typically show an external mixing state and size-dependent hygroscopicity (Zong et al., 2021). Thus, in order to capture the real and high-time-resolution features of ambient particles' hygroscopic growth in the HRT, direct particle hygroscopic growth measurements are a matter of necessity.**

**To our best knowledge, there are only limited studies exploring the impact of the hygroscopic growth of ambient particles on the particle deposition by direct hygroscopicity measurements. Moreover, hygroscopicity measurements using the Humidity Tandem Differential Mobility Analyzer (H-TDMA) (Londahl et al., 2009; Farkas et al., 2022; Vu et al., 2015; Kristensson et al., 2013) or Differential Aerosol Sizing and Hygroscopicity Spectrometer Probe (DASH-SP) (Youn et al., 2016) in these studies were all conducted at relatively lower RH (~ 90%) compared to that in the HRT (RH = 99.5%). For example, Farkas et al. (2022) modelled DFs of aerosol particles with four different diameters and studied in their dry state and after their hygroscopic growth at RH = 90% using a H-TDMA. Youn et al. (2016) examined**

size-resolved hygroscopicity data by DASH-SP for particles sampled near mining and smelting operations to study the effects of particles' hygroscopic growth on the HRT deposition of toxic contaminants. It was further assumed that κ was independent of RH on the premise that the effective molar volume of the solute does not vary with RH. However, the presence of surface active, slightly soluble substances, and the co-condensation of semi-volatile soluble organic compounds can result in the humidity-dependent characteristic of κ (Wu et al., 2013; Wex et al., 2009; Topping and Mcfiggans, 2012). For instance, Liu et al. (2018) showed that κ could vary from about 0.1 at RH < 20% to less than 0.05 when RH ≈ 90% due to the non-ideal mixing of water with hydrophobic and hydrophilic organic components. Therefore, an explicit hygroscopicity measurements at HRT-like conditions will make the deposition estimation more accurate."

Line 95-97 in the manuscript:

"The field campaign was conducted from June 8 to July 6 in 2014 at an ecological park in the rural area of Wangdu County (38.666ºN, 115.210ºE) in the North China Plain, a polluted area with high population density and strong primary emissions."

1. *Specific comments on the reply to review#1 to comment 3.*

*Response 1: The statement "only the comparison of κ measured simultaneously at the same sampling site makes sense." This is not what is the case. Why should the rural air masses at the measurement site be very different to other rural places. The κ value is a quantity commonly compared between different measurement sites. It only has to be clarified how the measurements were conducted and the RH or supersaturation need to be included into a comparison.*

[Response]: Thanks for the reviewer's comment. The result measured by HH-TDMA at RH = 98% in this study was similar to those measured by H-TDMA or HH-TDMA at RH = 87% - 98.5% in other rural sites in the North China Plain in summer. The related statement was added as follows.

[Revise]: Line 19-29 in the appendix:

"The particle hygroscopicity parameter (Kappa, κ) in this study and previous studies measured in rural sites in the North China Plain (NCP) was shown in Table S1. The average size-resolved κ was in the range of 0.24 - 0.32 during the sampling period. The hygroscopic properties of particles in this study was similar to those determined in the NCP in summer, such as in Wuqing (Liu et al., 2011) and Xianghe (Zhang et al., 2016), which was higher than that measured in winter, such as in Dingxing (Shi et al., 2022). It can be explained that the mass fraction of organic matters with relatively weak hygroscopicity was higher in winter, while secondary inorganic aerosols with strong hygroscopicity made higher contribution in summer (Sun et al., 2015). Besides, the particle hygroscopicity increased as the particle diameter increasing, which was in accordance with previous studies measured in urban and rural sites (Swietlicki et al., 2008)."

**Table S 1 Particle hygroscopicity parameter (Kappa, κ) in this study and previous studies**

| Rural site | Kappa, mean ± SD (Dry diameter, nm) | | | | | | Instrument | RH | Reference |
|---|---|---|---|---|---|---|---|---|---|
| Wangdu | 0.24 ± 0.09 (30) | 0.24 ± 0.07 (50) | 0.27 ± 0.06 (100) | 0.28 ± 0.07 (150) | 0.30 ± 0.08 (200) | 0.32 ± 0.10 (250) | HH-TDMA | 98% | This study |
| Wuqing | 0.25 ± 0.06 (50) | 0.28 ± 0.04 (100) | 0.33 ± 0.05 (200) | 0.35 ± 0.05 (250) | | | HH-TDMA | 98.5% | (Liu et al., 2011) |
| Xianghe | 0.29 ± 0.09 (50) | 0.30 ± 0.06 (100) | 0.31 ± 0.06 (150) | 0.33 ± 0.04 (200) | 0.35 ± 0.08 (250) | 0.37 ± 0.09 (350) | H-TDMA | 87% | (Zhang et al., 2016) |
| Dingxing | 0.16 (60) | 0.18 (100) | 0.16 (150) | 0.15 (200) | | | H-TDMA | 90% | (Shi et al., 2022) |

*2. Specific comments on the reply to review#1 to comment 14. L.363:*

*Response 2: What allows the conclusion that the measured BC is "pure BC"? It is an ambient measurement with many committed species. Thus, I would expect some aging of the BC particles. There are quite some studies focusing on the hygroscopicity of BC. E.g., Liu et al., 2013. https://acp.copernicus.org/articles/13/2015/2013/acp-13-2015-2013.pdf The statement "The κ of BC with Dp = 50 nm was 0.20 ± 0.12" should be rephrased. It is the κ of the aerosol population, not of BC. The mass ratio can be strongly influenced by relatively few larger particles, whereas κ depends on the whole size distribution.*

[Response]: Thanks for the reviewer's comment. As the expression was not rigorous, relevant statements have been deleted.

*3. Fig. S8 BC and hydrophobic particles cannot be used synonymous. There are also other hydrophobic particles. Please be clear.*

[Response]: Thanks for the reviewer's comment. The related statements and Figure S8 were deleted.

*Comments on the revised manuscript:*

*1. 253: Are there differences between Figure 2 and S2, S3? They look very comparable so a statement on the comparison will be nice.*

[Response]: Yes. Due to the similar physiological parameters (such as the FRC, URT volume, TV, and BF) of the adults (Figure 2 in the manuscript) and the elderly (Figure S3 in the SI), their regional and total deposition functions were similar to each other. While, for the reason that the FRC is positively correlated with the body weight, the FRC of the children was about one third of those of the adults and the elderly, leading to the different DF curves of the children (Figure S2 in the SI). Therefore, the statement of comparing the DFs of all age groups was added as follows.

**[Revise]: Line 244-252 in the manuscript:**

**"As shown in the three figures, the regional and total DFs of all age groups respectively followed the same trends regardless the particle hygroscopicity. Due to the similar physiological parameters (such as the FRC, URT volume, TV, and BF, as shown in Table 1) of the adults (Figure 2) and the elderly (Figure S3), their regional and total DF functions were similar to each other. While, for the reason that the FRC is positively correlated with the body weight, the FRC of the children was nearly one third of those of the adults and the elderly, which may lead to the different DF curves of the children (Figure S2). Compared with the adults and the**

**elderly, the children had lower DFs of ultrafine particles in the head and higher DFs of submicron particles in the P region, resulting in higher total DFs of submicron particles."**

*2. Figure 3: I recommend to merge fig. 3 and S5 to one figure. Figure 3 alone gives somehow misleading indications. The driving factor is the exposure time with not corresponding activities. As stated in the authors reply to the review, the activities for the exposure time are not only resting, it should be made clear that this figure represents the literature exposure time to ambient pollution for a certain region not for all activities but only for resting.*

[Response]: Thanks for the reviewer's suggestion. The former Figure S5 was incorporated into Figure 3. The related content was modified as follows.

**[Revise]: Line 184-189 in the manuscript:**

**"It should be noticed that the exposure time data came from the statistical results of the questionnaire survey of the outdoor activity time for the rural population in Hebei, China. While, people may rest, take light exercise, or take heavy exercise during the exposure time. Different exercise levels (e.g. sitting, walking, exercising, etc) can result in different dose estimations and are not discussed here. For instance, previous studies found that the exercise level had great impact on the minute ventilation and led to the increasing deposition dose (Londahl et al., 2007)."**

**Figure 3 and related statement were shown in Line 325-343 in the manuscript:**

[Figure]

**Figure 3. Regional and total deposition doses for the (a) children, (b) adults, and (c) elderly with/without considering particle hygroscopicity. The dark blue columns represent doses without considering hygroscopicity. The light blue columns**

represent doses considering hygroscopicity. The red lines on the column represent the division of doses of hygroscopic (above the red line) and hydrophobic particles (below the red line). Numbers above each column mean the corresponding particle doses with a unit of $10^{10}$ #/day. (d) The average regional and total deposition rates considering hygroscopicity for three age groups. The green, orange, and purple column represent the children, adults, and elderly, respectively.

"In both cases, the adults (Figure 3(b)) and the elderly (Figure 3(c)) groups received similar regional and total doses. In contrast, the children had the minimum total dose (Figure 3(a)), which was around half (47.4% on average) to that for the adults. As shown in Eq (4), the exposure time is an important parameter for deposition dose calculations. The exposure time of the adults and the elderly groups was more than twice than that of the children (Table 1), which resulted in the greater deposition dose for the former two groups. Therefore, to remove the impact of the exposure time, the regional and total deposition rates for three age groups were also calculated and shown in Figure 3(d). The children received the maximum total deposition rate (($4.81 \pm 4.55$) × $10^9$ #/h), followed by the elderly group (($4.09 \pm 3.92$) × $10^9$ #/h), and the adults received the minimum (($3.84 \pm 3.69$) × $10^9$ #/h). The regional deposition rate in the TB and P regions for three age groups showed a same order as the total deposition rate, while the order in the head was quite different. Specifically, three age groups had the similar deposition rate in the head."

*3. 453: There are already HHTDMA capable for measurements in nearly saturated conditions (up to 99.6% RH). E.g. Mikhailov and Vlasenko 2020. https://amt.copernicus.org/articles/13/2035/2020/*

[Response]: Thanks for the reviewer's comment. The statement was modified as follows.

**[Revise]: Line 437-443 in the manuscript:**

"For instance, the HH-TDMA (Suda and Petters, 2013), the Leipzig Aerosol Cloud Interaction Simulator (Stratmann et al., 2004), the inverted streamwise-gradient cloud condensation nuclei counter (Ruehl et al., 2010), and the filter-based differential hygroscopicity analyzer (Mikhailov et al., 2011) have been used to determine the particle hygroscopicity at RHs up to 99%. In particular, Mikhailov and Sergey (2020) adopted a new method with in situ restructuring to minimize the influence of particle shape, and the RH was up to 99.6% with an RH measurement accuracy better than 0.4%."

*4. 454: I recommend a broader literature research and to extend the literature review to other global regions. There are studies reporting physiological parameters in the HRT and may the authors would need to justify here why physiological parameters in the HRT are significantly different between some rural regions in China and other places.*

[Response]: Thanks for the reviewer's advice. We summarized the physiological parameters in the HRT used in the previous studies as Table R1.

**Table R 1 Physiological parameters of different populations**

| Groups | FRC / mL | URT Volume / mL | TV / mL | BF / min$^{-1}$ | Population | Reference |
|---|---|---|---|---|---|---|
| Children | 1330 | 21.91 | 630 | 22 | | |
| Adults | 3338 | 36.31 | 730 | 18 | Chinese | This study |
| Elderly | 3259 | 34.01 | 760 | 18 | | |
| Adults | 2950 | 44.70 | 537.5 | 16 | Chinese | (Li et al., 2016) |
| Children | 1330 | 22.79 | 580 | 32 | | |
| Adults | 3389 | 40.78 | 2360 | 26 | Greek | (Voliotis and Samara, 2018) |
| Elderly | 3475 | 34.63 | 2270 | 26 | | |
| Children | 1484 | 25 | 303 | 19 | | |
| Adults | 3122 | 50 | 625 | 16 | Chinese | (Wang et al., 2021) |
| Elderly | 3402 | 50 | 625 | 16 | | |
| Adults | 3300 | 50 | 1250 | 20 | American | (Gangwal et al., 2011) |
| Adults | 3300 | 50 | 750 | 12 | Caucasian | (Manigrasso et al., 2015) |
| Adults | 3301 | 50 | 750 | 12 | Caucasian | (Vu et al., 2017) |
| General | 3300 | 50 | 625 | 12 | - | MPPD Help |

As shown in Table R1, the physiological parameters used in this study are similar to those for Chinese in other studies and those for other populations.

*Anonymous Referee #2*

*Thank you for the changes and responses to my suggestions and questions. The only point I think still needs to be addressed is the question of deposited dose. I understand that determining dose on a mass basis is not possible (could it be estimated?). Thus, in the abstract, I think it is important to note that the doses change on a particle number basis to avoid any misunderstanding.*

[Response]: Thanks for the reviewer's suggestion. The statement was modified in the manuscript.

[Revise]: Line 19-23 in the manuscript:

[revised manuscript text omitted]